# RayDF: Neural Ray-surface Distance Fields with Multi-view Consistency

**Zhuoman Liu**    **Bo Yang**∗    **Yan Luximon**    **Ajay Kumar**    **Jinxi Li**
vLAR Group, The Hong Kong Polytechnic University
{zhuo-man.liu, jinxi.li}@connect.polyu.hk    bo.yang@polyu.edu.hk

## Abstract

In this paper, we study the problem of continuous 3D shape representations. The majority of existing successful methods are coordinate-based implicit neural representations. However, they are inefficient to render novel views or recover explicit surface points. A few works start to formulate 3D shapes as ray-based neural functions, but the learned structures are inferior due to the lack of multi-view geometry consistency. To tackle these challenges, we propose a new framework called **RayDF**. It consists of three major components: 1) the simple ray-surface distance field, 2) the novel dual-ray visibility classifier, and 3) a multi-view consistency optimization module to drive the learned ray-surface distances to be multi-view geometry consistent. We extensively evaluate our method on three public datasets, demonstrating remarkable performance in 3D surface point reconstruction on both synthetic and challenging real-world 3D scenes, clearly surpassing existing coordinate-based and ray-based baselines. Most notably, our method achieves a $1000\times$ faster speed than coordinate-based methods to render an $800 \times 800$ depth image, showing the superiority of our method for 3D shape representation. Our code and data are available at https://github.com/vLAR-group/RayDF

## 1 Introduction

Learning accurate and efficient 3D shape representations is crucial for many cutting-edge applications in the fields of machine vision and robotics. Recent advances in 3D coordinate-based neural representations including occupancy fields (OF) [46, 11], un/signed distance fields (U/SDF) [54, 14], and radiance fields (NeRF) [47], have shown great potential to recover complex shapes from RGB/D images and/or point clouds. Although these methods and their variants have achieved excellent performance in a wide range of downstream tasks such as shape reconstruction [55, 58, 60], novel view synthesis [4, 72, 51, 5], and scene understanding [85, 86, 73], obtaining 3D shapes or 2D views from their trained networks is computationally expensive, due to the requirement of extensive network evaluations to regress surface points.

Very recently, a number of works start to represent 3D models as ray-based neural functions. By simply taking individual light rays as input, the methods LFN [64] and NeuLF [40] learn to directly predict the radiance (RGB) values, while PRIF [23] and DDF [3] straightly estimate the surface hitting points of input rays. Compared with coordinate-based representations, these ray-based methods are clearly more efficient to infer 3D geometry and render 2D views, as every single ray only needs to query the trained network once forward. Nevertheless, the learned 3D geometries by these methods are still lack of fidelity, primarily because they fail to explicitly take into account geometry consistency across multiple views, resulting in the network likely over-fitting individual training rays but unable to generalize to unseen rays in testing.

In this paper, we aim to address this key issue of ray-based neural representations by explicitly integrating multi-view geometry consistency into the network design. In particular, we introduce a

---

∗Corresponding Author

37th Conference on Neural Information Processing Systems (NeurIPS 2023).

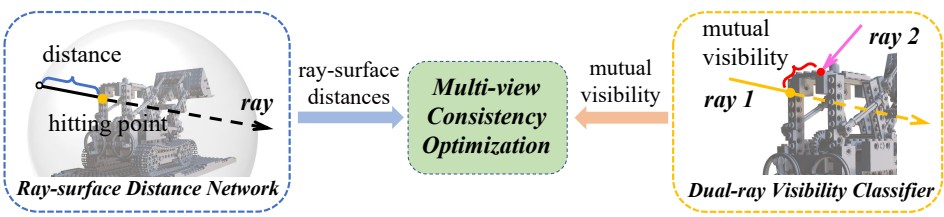

Figure 1: The general workflow and components of our framework.

new pipeline as shown in Figure 1. It consists of two independent neural networks together with a particular training module: 1) the main ray-surface distance network, 2) an auxiliary dual-ray visibility classifier, and 3) a multi-view consistency optimization module.

The **main network** simply takes a ray as input and directly infers the distance between ray origin and its hitting point on the surface. Basically, this network elegantly represents 3D shapes as ray-based implicit functions, denoted as *ray-surface distance fields*, keeping the unique advantage in efficiency for shape extraction and rendering. The **auxiliary network** takes a pair of rays as input and predicts their mutual visibility. In fact, this is a simple binary classifier, aiming at distinguishing whether any two rays hit at the same surface point or not, *i.e.*, the mutual visibility. Having a trained auxiliary network at hand, the **multi-view consistency optimization module** specifies how to effectively leverage the learned dual-ray visibility to train the main network, thus driving learned ray-surface distances to be multi-view consistent from any seen or unseen viewing angles.

Since our ray-surface distance fields primarily aim at representing accurate 3D shapes, the whole pipeline is designed to be trained on depth images, although light fields (radiance) can be optionally learned in parallel if color images are also available in training. In this regard, the closest works to ours are PRIF [23] and DDF [3], neither of which explicitly considers the multi-view geometry consistency. Overall, compared with all existing coordinate-based representations, our method keeps the superiority in efficiency thanks to the ray-based formulation. Compared with the existing ray-based approaches, ours excels at learning accurate 3D geometries thanks to the multi-view consistency for **ray**-surface **d**istance **f**ields. Our method is called **RayDF** and our contributions are:

- We employ a straightforward ray-surface distance field for representing 3D shapes. This formulation is inherently more efficient than existing coordinate-based representations.
- We design a new dual-ray visibility classifier to learn the spatial relationships of any pair of rays, enabling the learned ray-surface distance fields to be multi-view geometry consistent.
- We demonstrate superior 3D shape reconstruction accuracy and efficiency on multiple datasets, showing significantly better results than existing coordinate-based and ray-based baselines.

## 2   Related Work

**Explicit 3D Shape Representations:** Classic methods to recover explicit 3D geometry of objects and scenes mainly include SfM [53] and SLAM [7]systems such as Colmap [62] and ORB-SLAM [49]. Another classic method is space carving [45, 37], which involves the process of carving out the voxel grid by utilizing multiple distinct views to obtain a robust approximation within the voxel space. To model explicit 3D structures, deep learning-based methods have shown impressive progress in recovering voxel grids [15, 80, 81], point clouds [21], octree [71], polygon meshes [34, 26] and shape primitives [87]. A comprehensive survey of these methods can be found in [6, 28]. Thanks to the large-scale datasets [10] for training sophisticated neural networks [29, 31], these methods demonstrate excellent results in many downstream tasks such as shape reconstruction [76, 70], generation [38], and semantic scene perception [67, 17, 25]. However, the fidelity of these discrete shape representations is primarily limited by their spatial resolutions and memory footprint.

**Coordinate-based Implicit 3D Shape Representations:** To avoid the discretization issue of traditional 3D representations, there has been a growing interest in developing implicit neural 3D representations using simple MLPs to represent continuous 3D shapes. Inspired by the seminal *coordinate-based* methods [11, 46, 54] which have shown great success in encoding high-quality 3D shapes, a plethora of follow-up works have been developed to tackle various vision tasks in [65, 79, 2, 12, 13, 20, 33, 43, 42, 27, 83]. These coordinate-based representations can be generally classified as: 1) occupancy fields (OF) [46, 11], 2) signed distance fields (SDF) [54], 3) unsigned

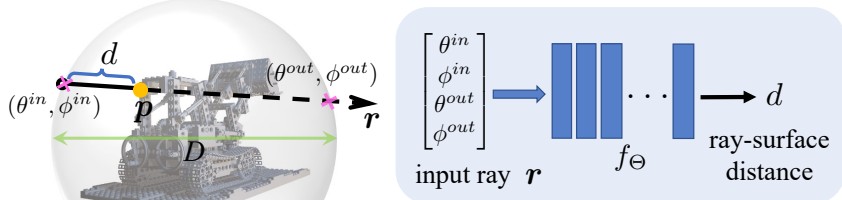

Figure 2: The left block illustrates the spherical parameterization of an input ray $\boldsymbol{r}$, and the right block shows our simple MLP-based ray-surface distance field $f_\Theta$.

distance fields (NDF) [14, 74], and 4) radiance fields (NeRF) [47]. More related methods can be found in surveys [24, 18]. Among them, the OF/SDF/NDF based methods demonstrate exceptional accuracy in recovering continuous 3D structures thanks to the simple level-set formulation, while the NeRF based methods achieve an unprecedented level of fidelity in rendering 2D views thanks to the successful volume rendering equation. However, all these coordinate-based representations inevitably require dense 3D location sampling and evaluations on trained networks to regress explicit surface points, though advanced techniques [41, 57, 69, 39, 30, 56, 50, 82, 48] can mitigate this issue somewhat but at the expense of a large memory increase. In this paper, our RayDF only needs a single network evaluation to regress every surface point, while still achieving on par or better reconstruction accuracy with existing OF/SDF/NDF methods.

**Ray-based Implicit 3D Shape Representations:** To overcome the inefficiency of coordinate-based 3D representations, a number of works start to formulate 3D shapes as ray-based neural functions via MLPs. Ray-based methods simply take individual rays as input and directly output either radiance values (RGB), *i.e. light fields*, or surface points, *i.e. distance fields*. LFN [64] and NeuLF [40] are among the early works of light fields to encode 3D representations from observed color images. Other light field works include [22, 61, 78, 52, 68, 75, 9, 1]. PRIF [23], DDF [3] and DRDF [36] are the early ray-based distance fields to learn 3D shapes from observed depth scans. Although these methods have shown very encouraging results for novel view rendering or surface reconstruction, they are usually limited to datasets with small baselines across multi-views. Basically, this is because the multi-view geometry consistency is not taken into account in their network designs. By contrast, in our RayDF pipeline, we explicitly introduce a dual-ray visibility classifier to aid our ray-surface distance fields to be multi-view shape consistent during training.

## 3 RayDF

### 3.1 Overview

Our pipeline consists of two networks and an optimization module. As shown in Figure 2, for the main network: **ray-surface distance field** $f_\Theta$, it takes a single oriented ray $\boldsymbol{r}$ as input, and directly regresses the distance $d$ between ray starting point and its surface hitting point. For the input ray $\boldsymbol{r}$, we opt for the conventional spherical parameterization [32], as it supports querying from $360°$ viewing angles. In particular, a fix-sized sphere is predefined with a relatively large diameter $D$ as a convex hull in which the target 3D scene is bounded. Each surface point $\boldsymbol{p}$ of the target scene can be regarded as a directional ray penetrating the sphere with two intersection points on the sphere. Each intersection point is parameterized by two variables specifying the angles with regard to the sphere center. Then, for each ray, we have a 4D parameterization $\boldsymbol{r} = (\theta^{in}, \phi^{in}, \theta^{out}, \phi^{out})$. Formally, the ray-surface distance field is defined below and implementation details are in Appendix A.1.

$$d = f_\Theta(\boldsymbol{r}), \quad \text{where } \boldsymbol{r} = (\theta^{in}, \phi^{in}, \theta^{out}, \phi^{out}), \quad \Theta \text{ are trainable parameters of MLPs} \quad (1)$$

For the auxiliary network: **dual-ray visibility classifier** $h_\Phi$, it takes a pair of rays as input and simply predicts their mutual visibility, aiming at explicitly modeling the mutual spatial relationships between any pairs of rays. This network, once well-trained, will play a key role in the third component: **multi-view consistency optimization**. Detailed designs are discussed in Sections 3.2 & 3.3.

### 3.2 Dual-ray Visibility Classifier

The main ray-surface network alone can indeed well fit many training ray-distance pairs, but there is no mechanism driving its output distances, *i.e.*, surface geometry, to be consistent across multiple views, especially for unseen views. For example, as illustrated in the leftmost block of Figure 3, if

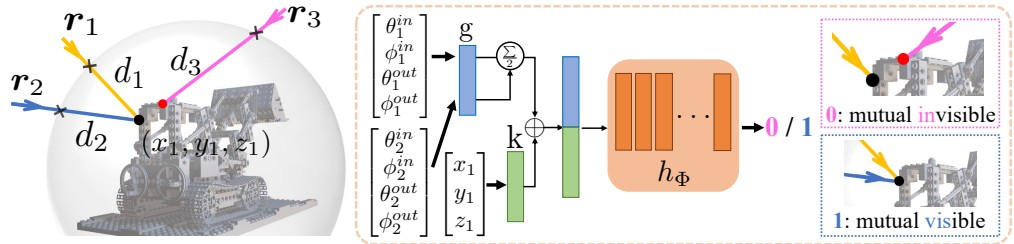

Figure 3: The leftmost block illustrates the mutual visibility of a pair of rays. The remaining block shows the symmetric design of our dual-ray visibility classifier $h_\Phi$.

two input rays $\boldsymbol{r}_1$ and $\boldsymbol{r}_2$ are mutually visible in the scene, both rays must hit at the same surface point, then the corresponding ray-surface distances $d_1$ and $d_2$ must satisfy a transformation equation $\boldsymbol{r}_1^{in} + d_1\boldsymbol{r}_1^{d} = \begin{pmatrix} x_1 \\ y_1 \\ z_1 \end{pmatrix} = \boldsymbol{r}_2^{in} + d_2\boldsymbol{r}_2^{d}$, where $\boldsymbol{r}^{d} = \frac{\boldsymbol{r}^{out} - \boldsymbol{r}^{in}}{\|\boldsymbol{r}^{out} - \boldsymbol{r}^{in}\|}$ and $\boldsymbol{r}^* = \begin{pmatrix} \sin\theta^*\cos\phi^* \\ \sin\theta^*\sin\phi^* \\ \cos\phi^* \end{pmatrix}$, $* \in \{in, out\}$ (More details in Section A.1.3), according to both rays' angles. Similarly, if input rays $\boldsymbol{r}_1$ and $\boldsymbol{r}_3$ are mutually invisible, then they should never satisfy the transformation equation because both never meet at the same surface point. From this fundamental principle, we can see that the mutual visibility between any two rays is crucial to inform the ray-surface distance network to be multi-view consistent.

To this end, we design a binary classifier to discriminate the visibility of an input pair of rays. A naïve idea is to simply concat two rays as input, feeding into MLPs to regress 0/1, where 1 represents *visible* and 0 otherwise. However, such a design fails to retain the symmetry of two rays. Here symmetry means that the visibility of two rays must be invariant to the input order of two rays. With this point, as illustrated in the right block of Figure 3, our visibility classifier $h_\Phi$ is formally defined as:

$$h_\Phi: \quad MLPs\left[\frac{g(\theta_1^{in}, \phi_1^{in}, \theta_1^{out}, \phi_1^{out}) + g(\theta_2^{in}, \phi_2^{in}, \theta_2^{out}, \phi_2^{out})}{2} \oplus k(x_1, y_1, z_1)\right] \to 0/1 \quad (2)$$

where $(\theta_1^{in}, \phi_1^{in}, \theta_1^{out}, \phi_1^{out})$ and $(\theta_2^{in}, \phi_2^{in}, \theta_2^{out}, \phi_2^{out})$ are the parameterizations of two input rays $\boldsymbol{r}_1$ and $\boldsymbol{r}_2$ respectively; $g()$ is a shared single fully-connected layer followed by an average pooling, thus guaranteeing the symmetry of input rays. Note that, the surface hitting point of $\boldsymbol{r}_1$, *i.e.*, $(x_1, y_1, z_1)$, is also encoded via a separate single fully-connected layer $k()$, followed by a concatenation operation denoted by $\oplus$, thus enhancing the pooled ray features, as we empirically find that such an enhancement could notably improve the classifier's accuracy. Implementation details are in Appendix A.1.

## 3.3 Multi-view Consistency Optimization

With the designed main ray-surface distance network $f_\Theta$ and the auxiliary dual-ray visibility classifier $h_\Phi$ at hand, we introduce the crucial multi-view consistency optimization module to train both networks. In particular, given $K$ posed depth images ($H \times W$) of a static 3D scene as the whole training data, our training module consists of two stages.

### Stage 1 - Training Dual-ray Visibility Classifier

The key to training this classifier is to create correct data pairs. First of all, all raw depth values are converted to ray-surface distance values. For a specific $i^{th}$ ray (pixel) in the $k^{th}$ image, we project its ray-surface point back to the remaining $(K-1)$ scans, obtaining the corresponding $(K-1)$ distance values. We set 10 millimeters as the *closeness* threshold to determine whether the projected $(K-1)$ rays are visible in the $(K-1)$ images. In total, we generate $K * H * W * (K-1)$ pairs of rays together with 0/1 labels. The standard cross-entropy loss function is adopted to optimize our dual-ray visibility classifier. More details on training data generation and implementation are in Appendix A.1.3.

Note that, this classifier is trained in a scene-specific fashion. Once the network is well-trained, it basically encodes the relationships between any two rays of a specific scene into network weights.

### Stage 2 - Training Ray-surface Distance Network

The ultimate goal of our whole pipeline is to optimize the main ray-surface network and drive it to be multi-view geometry consistent. However, this is non-trivial as simply fitting the network with ray-surface data points cannot generalize to unseen rays, which can be seen in our ablation study in Section 4.5. In this regard, we fully leverage the well-trained visibility classifier to aid our training of ray-surface distance network. Particularly, this stage consists of the following key steps:

- Step 1: All depth images are converted to ray-surface distances, generating $K * H * W$ training ray-distance pairs for a specific 3D scene.
- Step 2: As illustrated in Figure 4, for a specific training ray $(\boldsymbol{r}, d)$, called *primary ray*, we uniformly sample $M$ rays $\{\boldsymbol{r}^1 \cdots \boldsymbol{r}^m \cdots \boldsymbol{r}^M\}$, called *multi-view rays*, in a ball centering at the surface point $\boldsymbol{p}$. We then calculate the distance between surface point $\boldsymbol{p}$ and the bounding sphere along each of $M$ rays, obtaining multi-view distances $\{\tilde{d}^1 \cdots \tilde{d}^m \cdots \tilde{d}^M\}$. This can be easily achieved according to the given distance $d$ in the training set. $M$ is simply set as 20 and more details are in Appendix A.4.
- Step 3: We establish $M$ pairs of rays $\{(\boldsymbol{r}, \boldsymbol{p}, \boldsymbol{r}^1) \cdots (\boldsymbol{r}, \boldsymbol{p}, \boldsymbol{r}^m) \cdots (\boldsymbol{r}, \boldsymbol{p}, \boldsymbol{r}^M)\}$ and then feed them into the well-trained visibility classifier $h_\Phi$, inferring their visibility scores $\{v^1 \cdots v^m \cdots v^M\}$.
- Step 4: We feed the primary ray and all sampled $M$ multi-view rays $\{\boldsymbol{r}, \boldsymbol{r}^1 \cdots \boldsymbol{r}^m \cdots \boldsymbol{r}^M\}$ into the ray-surface distance network $f_\Theta$, estimating their surface distances $\{\hat{d}, \hat{d}^1 \cdots \hat{d}^m \cdots \hat{d}^M\}$. Since the network $f_\Theta$ is randomly initialized, thus the estimated distances are inaccurate in the beginning.
- Step 5: We design the following multi-view consistency loss function to optimize the ray-surface distance network until convergence:

$$\ell_{mv} = \frac{1}{\sum_{m=1}^M v^m + 1} \left( |\hat{d} - d| + \sum_{m=1}^M \left( |\hat{d}^m - \tilde{d}^m| * v^m \right) \right) \tag{3}$$

Basically, this simple loss drives the network to not only fit the primary ray-surface distance (seen rays in the training set), but also satisfy that the visible multi-view rays (unlimited unseen rays in the training set) also have accurate distance estimations.

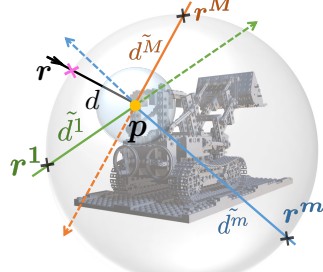

Figure 4: Multi-view ray sampling.

### 3.4 Surface Normal Derivation and Outlier Points Removal

In the above Sections 3.1&3.2&3.3, we have two network designs and an optimization module to train them separately. Nevertheless, we empirically find that the main ray-surface distance network may predict inaccurate distance values, particularly for rays near sharp edges. Essentially, this is because the actual ray-surface distances may be discontinuous at sharp edges given extreme viewing angle changes. This shape discontinuity is actually a common challenge for almost all existing implicit neural representations, because modern neural networks can only model continuous functions in theory.

Fortunately, a nice property of our ray-surface distance field is that the normal vector at every estimated 3D surface point can be easily derived in a closed-form expression using auto differentiation of the network. In particular, given an input ray $\boldsymbol{r} = (\theta^{in}, \phi^{in}, \theta^{out}, \phi^{out})$, and its estimated ray-surface distance $\hat{d}$ from network $f_\Theta$, the corresponding normal vector $\boldsymbol{n}$ of that estimated surface point can be derived as a specific function shown below.

$$\boldsymbol{n} = Q\left(\frac{\partial \hat{d}}{\partial \boldsymbol{r}}, \boldsymbol{r}, D\right), \text{ details of the function } Q \text{ and derivation are in Appendix A.2.} \tag{4}$$

With this normal vector, we may choose to add an additional loss to regularize the estimated surface points to be as smooth as possible. Yet, we empirically find that the overall performance improvement on an entire 3D scene is rather limited, as these extremely discontinuous cases are actually sparse.

In this regard, we turn to simply removing the predicted surface points, *i.e.*, outliers, whose normal vectors' Euclidean distances are larger than a threshold in the network inference stage. In fact, PRIF [23] also adopts a similar strategy to filter out outliers. Note that, advanced smoothing or interpolating techniques may be integrated to improve our framework, which is left for future exploration.

## 4 Experiments

Our method is evaluated on three types of public datasets: 1) the object-level synthetic Blender dataset from the original NeRF paper [47], 2) the scene-level synthetic DM-SR dataset from the recent DM-NeRF paper [73], and 3) the scene-level real-world ScanNet dataset [16].

**Baselines:** We carefully select the following six successful and representative implicit neural shape representations as our baselines: 1) OF [46], 2) DeepSDF [54], 3) NDF [14], 4) NeuS [77], 5) DS-NeRF [19], 6) LFN [64], and 7) PRIF [23]. The OF/DeepSDF/NDF/NeuS methods are coordinate-based level-set methods, showing outstanding performance in modeling 3D structures. DS-NeRF

is a depth-supervised NeRF [47], inheriting excellent capability in rendering 2D views. LFN and PRIF are two ray-based methods with superior efficiency in generating 2D views. We note that there are many sophisticated variants of these baselines, achieving SOTA performance on various datasets. Nevertheless, we do not intend to comprehensively compare with them, basically because many of their techniques such as more advanced implementations, adding additional conditions, replacing with more powerful backbones, *etc.*, can be easily integrated into our framework as well. We leave these potential improvements for future exploration but only focus on our vanilla ray-surface distance field in this paper. For a fair comparison, all baselines are supervised with the same amount of depth scans as ours, carefully trained from scratch in the same scene-specific fashion. More details about the implementation of all baselines and possible minor adaptations are in Appendix A.3.1.

**Metrics:** For evaluation metrics of shape reconstruction, we report: 1) the per ray-surface **absolute distance error (ADE)** in centimeters averaged across all testing images, 2) the **Chamfer distance (CD)** [21] between the whole reconstructed structure and the ground truth shape for each scene. As to our method, we use the existing TSDF fusion [84] to obtain a full mesh from predicted depths at the testing views, on which we uniformly sample 30K points. Another 30K points are uniformly sampled from the ground truth 3D mesh for calculating CD. The outlier point removal is only applied to clean the reconstructed point clouds when calculating CD. No additional post-processing steps are employed when computing all ADE scores. Apparently, ADE is more accurate to evaluate the surface estimation of our method, because the CD scores may be biased due to the extra TSDF fusion. In this regard, the CD scores are just presented as a complementary metric to show the general shape quality in the whole 3D scene space. More details about how to obtain the reconstructed 3D full shape for each baseline and our method are in Appendix A.3.2. For evaluation metrics of appearance reconstruction, *i.e.* novel view synthesis of 2D color images, the standard **PSNR**, **SSIM**, and **LPIPS** scores are reported following NeRF [47]. We present **Accuracy** (%) and **F1-Score** (%) as evaluation metrics of the dual-ray visibility classifier. For qualitative results, refer to Figure 5 and Appendix A.7.

## 4.1 Efficiency of RayDF

Similar to the existing ray-based methods LFN and PRIF, our RayDF also has the superiority in efficiency to generate 2D images or recover the explicit surface points. To quantitatively compare the efficiency with baselines, we conduct a simple experiment to generate an $800 \times 800$ depth image on a computer with a single NVIDIA RTX 3090 GPU card and a CPU of AMD Ryzen 7. As shown in Table 1, it can be seen that, not surprisingly, the coordinate-based methods OF/ DeepSDF/ NDF/ NeuS/ DS-NeRF are extremely slow to render a high-resolution depth image. In particular, OF [46] needs a large number of small steps to gradually approach the surface point along a given light ray direction, while DeepSDF [54] and NDF [14] rely on expensive sphere tracing to

Table 1: Rendering time consumption (seconds).

|  | Time |
|---|---|
| OF [46] | 286.057 |
| DeepSDF [54] | 17.590 |
| NDF [14] | 28.310 |
| NeuS [77] | 32.793 |
| DS-NeRF [19] | 25.612 |
| LFN [64] | 0.017 |
| PRIF [23] | **0.013** |
| **RayDF (Ours)** | 0.019 |

regress the points. NeuS [77] and DS-NeRF [19] need extra post-processing steps to calculate depth values from densities. By contrast, our RayDF and the existing ray-based methods achieve more than $1000\times$ faster speed to render a dense depth view.

## 4.2 Evaluation on Blender Dataset

The Blender dataset from NeRF [47] consists of pathtraced images of 8 synthetic objects with complicated geometry and realistic materials. Each object has 100 views for training and 200 novel views for testing. Each image has $800 \times 800$ pixels. Since the released dataset does not include depth images, we just use the provided Blender files to generate additional depth scans of the same resolution exactly following the original 2D view poses. Note that, the physical sizes of these models are about 2.5 meters in length/height/width, so the sphere diameter $D$ is set as 3 meters in our method. For the baselines and our method, we conduct the following two groups of experiments.

- **Group 1 - 3D Shape Representation Only:** Since our RayDF needs depth scans in training to learn continuous 3D shape representations, we conduct this group of experiments only on multi-view depth images. All baselines are also trained on the same number of depth views.
- **Group 2 - 3D Shape and Appearance Representations:** Our RayDF is also flexible to add a parallel branch to output radiance field, *i.e.*, RGB values, so that the continuous appearance representations can be simultaneously learned. In this regard, we conduct this group of experiments on both RGB images and depth images. The baselines NeuS/ DS-NeRF/ LFN/ PRIF are also trained on the same RGBD images for comparison. The other three methods OF/ DeepSDF/ NDF are not

Table 2: Quantitative results of all baselines and our method in experiments of Groups 1&2. All scores are averaged out across 8 scenes of the Blender dataset [47].

| | Group 1 | | Group 2 | | | | |
| --- | --- | --- | --- | --- | --- | --- | --- |
| | ADE↓ | CD ($\times 10^{-3}$)↓ mean / median | ADE↓ | CD ($\times 10^{-3}$)↓ mean / median | PSNR↑ | SSIM↑ | LPIPS↓ |
| OF [46] | 10.57 | 2.982 / 0.706 | - | - | - | - | - |
| DeepSDF [54] | 12.95 | 3.382 / 0.679 | - | - | - | - | - |
| NDF [14] | 12.14 | **2.976** / 0.831 | - | - | - | - | - |
| NeuS [77] | 11.88 | 4.756 / 0.907 | 12.10 | 4.662 / 0.938 | **27.19** | 0.910 | 0.100 |
| DS-NeRF [19] | 13.22 | 117.270 / 1.135 | 14.64 | 143.295 / 1.760 | 26.63 | **0.933** | **0.063** |
| LFN [64] | 24.47 | 89.425 / 15.681 | 12.33 | 60.289 / 1.230 | 23.20 | 0.888 | 0.125 |
| PRIF [23] | 14.68 | 20.764 / 1.677 | 14.56 | 21.279 / 1.693 | 23.31 | 0.874 | 0.152 |
| **RayDF (Ours)** | **7.97** | 3.388 / **0.663** | **8.17** | **3.295 / 0.755** | 26.52 | 0.910 | 0.099 |

included here, because it is non-trivial to add RGB supervision due to their level-set formulation. Details of the parallel branch for all methods are in Appendix A.3.

**Analysis:** Table 2 shows the quantitative comparison. We can see that: 1) Our RayDF achieves significantly better results for explicit surface recovering on both groups of experiments, especially on the most important ADE metric, demonstrating the clear superiority over both coordinate and ray based baselines. 2) Our method also achieves comparable performance with DS-NeRF for novel view synthesis, being better than LFN and PRIF and showing the flexibility of our framework.

## 4.3 Evaluation on DM-SR Dataset

The DM-SR dataset from the recent DM-NeRF paper [73] consists of 8 synthetic complex 3D indoor rooms. The room types and designs follow Hypersim dataset [59], and the rendering trajectories for both training and testing images follow the Blender dataset of NeRF [47]. For each 3D scene, there are 300 views for training and 100 novel views for testing. Each view has $400 \times 400$ pixels. Each scene has a physical size of about 10 meters in length/height/width, so the sphere diameter $D$ is set as 11 meters in our method. Similarly, we conduct the following two groups of experiments.

- **Group 1 - 3D Shape Representation Only:** We conduct this group of experiments only on multi-view depth images for all methods.
- **Group 2 - 3D Shape and Appearance Representations:** We conduct this group of experiments on both RGB images and depth images for NeuS/ DS-NeRF/ LFN/ PRIF and our method.

Table 3: Quantitative results of all baselines and our method in experiments of Groups 1&2. All scores are averaged out across 8 scenes of the DM-SR dataset [73].

| | Group 1 | | Group 2 | | | | |
| --- | --- | --- | --- | --- | --- | --- | --- |
| | ADE↓ | CD ($\times 10^{-3}$)↓ mean / median | ADE↓ | CD ($\times 10^{-3}$)↓ mean / median | PSNR↑ | SSIM↑ | LPIPS↓ |
| OF [46] | 15.83 | 11.402 / 4.888 | - | - | - | - | - |
| DeepSDF [54] | 16.97 | **11.281** / 5.087 | - | - | - | - | - |
| NDF [14] | 22.41 | 12.300 / 5.911 | - | - | - | - | - |
| NeuS [77] | 9.94 | 12.744 / **4.620** | 11.66 | 15.017 / 5.308 | **33.22** | 0.965 | 0.054 |
| DS-NeRF [19] | 10.77 | 25.380 / 6.032 | 10.77 | 25.548 / 6.102 | 31.83 | **0.977** | **0.026** |
| LFN [64] | 18.30 | 13.673 / 5.372 | 18.51 | **14.085** / 5.349 | 30.86 | 0.930 | 0.111 |
| PRIF [23] | 11.89 | 25.993 / 5.159 | 11.77 | 24.842 / **5.156** | 31.01 | 0.933 | 0.111 |
| **RayDF (Ours)** | **7.41** | 14.272 / 5.353 | **7.97** | 14.251 / 5.300 | 30.32 | 0.940 | 0.113 |

**Analysis:** From Table 3, it can be seen that our method again surpasses all baselines on the most critical ADE metric in both groups of experiments, though the rough metric CD scores are just comparable with baselines potentially due to the inaccuracy incurred by external TSDF fusion. In the meantime, our method still obtains high-quality novel view synthesis, without noticeably sacrificing shape representation when trained with both RGB images and depth scans in Group 2.

## 4.4 Evaluation on ScanNet Dataset

We further evaluate our method on the challenging real-world 3D dataset ScanNet [16]. We randomly select 6 scenes for evaluation. Originally, each scene has more than 2000 RGBD scans in a relatively

Table 4: Quantitative results of all baselines and our method in experiments of Groups 1&2. All scores are averaged out across 6 scenes of the ScanNet dataset [16].

| | Group 1 | | Group 2 | | | | |
| | ADE↓ | CD ($\times 10^{-3}$)↓ mean / median | ADE↓ | CD ($\times 10^{-3}$)↓ mean / median | PSNR↑ | SSIM ↑ | LPIPS ↓ |
|---|---|---|---|---|---|---|---|
| OF [46] | 15.84 | 8.991 / 2.810 | - | - | - | - | - |
| DeepSDF [54] | 12.07 | 17.200 / 6.255 | - | - | - | - | - |
| NDF [14] | 22.15 | 30.436 / 8.348 | - | - | - | - | - |
| NeuS [77] | 17.20 | 30.475 / 3.771 | 24.12 | 67.504 / 5.658 | 27.00 | 0.817 | 0.253 |
| DS-NeRF [19] | 6.64 | 9.653 / 2.318 | 7.84 | 14.642 / 2.950 | 25.03 | 0.855 | **0.197** |
| LFN [64] | 6.53 | **7.651** / 1.864 | 6.77 | 31.928 / 2.162 | 28.14 | 0.814 | 0.262 |
| PRIF [23] | 10.32 | 19.662 / 2.828 | 10.65 | 22.902 / 2.804 | 21.14 | 0.731 | 0.388 |
| **RayDF (Ours)** | **5.42** | 10.517 / **1.848** | **5.31** | **10.739 / 2.009** | **31.58** | **0.856** | 0.224 |

long sequence, whose camera poses are usually facing outwards within indoor rooms. Such long camera trajectories pose a great challenge to our method due to our sphere parameterization of input rays. To overcome this issue, we take the divide and conquer strategy to process a subset of the full video frames for each scene. In particular, for each scene, we only pick up a subset of 300 continuous RGBD scans in our experiment, where 200 are uniformly sampled for training, and 100 for testing. As to these partial 3D scenes, the physical sizes are between $4 \sim 16$ meters in length/height/width, so the sphere diameter $D$ is set as 16 meters in our method. Similarly, we conduct the following two groups of experiments.

- **Group 1 - 3D Shape Representation Only:** We conduct this group of experiments only on multi-view depth images for all methods.
- **Group 2 - 3D Shape and Appearance Representations:** We conduct this group of experiments on both RGB images and depth images for NeuS/ DS-NeRF/ LFN/ PRIF and our method.

**Analysis:** Table 4 compared all methods on the challenging real-world scenes. Remarkably, our RayDF clearly outperforms all baselines in almost all evaluation metrics in both groups of experiments, showing a distinct advantage in representing noisy and complex real-world 3D scenes. This confirms that multi-view geometry consistency is indeed essential in learning 3D representations.

### 4.5 Ablation Study

Since our framework just has two MLP-based neural networks, there is few hyperparameter to tune. The key component here is the dual-ray visibility classifier. To evaluate its effectiveness, we mainly conduct the following ablations on the classifier on Blender dataset [47] only with depth images.

**(1) Removing the dual-ray visibility classifier:** Without the classifier, the multi-view consistency loss function becomes the simple $\ell_1$ loss: $\ell_{mv} = |\hat{d} - d|$ for rays in the training depth scans.
**(2) Replacing position $(x, y, z)$ by ray distance $d$ in function $k()$:** This is to evaluate the enhancement of explicitly adding surface point position.
**(3) Removing the function $k()$ and only using two rays:** This ablation is to demonstrate the necessity of adding surface information for accurate classification.
**(4) Concatenating two rays instead of average pooling:** This ablation aims to evaluate the effect of the symmetry of the input rays on our classifier.
**(5) Adding noises to the well-trained visibility classifier:** Various levels of noise are introduced to the well-trained classifier to assess the reliability of our visibility classifier. In particular, we directly add a random noise drawn from a normal distribution $(0, \sigma^2)$ to the visibility score with a clip between 0 and 1, and then use the noisy score in our multi-view consistency loss $\ell_{mv}$ in Eq. 3 to optimize our ray surface distance network.

Table 5: Ablation studies on the classifier.

| | Acc.↑ | F1↑ | ADE↓ | CD ($\times 10^{-3}$) ↓ mean / median |
|---|---|---|---|---|
| (1) w/o classifier $h_\Phi$ | - | - | 47.08 | 1328.934 / 1073.704 |
| (2) use $d$ in function $k$ | 81.56 | 77.67 | 10.82 | 4.984 / 0.994 |
| (3) remove function $k$ | 81.93 | 76.77 | 10.23 | 4.753 / 0.890 |
| (4) use $g(\boldsymbol{r}_1) \oplus g(\boldsymbol{r}_2)$ | **95.06** | 50.17 | 9.77 | 4.740 / 0.825 |
| **Full RayDF** | 86.34 | **84.23** | **7.97** | **3.388 / 0.663** |

Table 6: Ablation study on the classifier with noises.

| Noise level ($\sigma^2$) | Acc.↑ | F1↑ | ADE↓ | CD ($\times 10^{-3}$) ↓ mean / median |
|---|---|---|---|---|
| 1.0 | 62.51 | 61.46 | 17.03 | 4.032 / 0.935 |
| 0.5 | 66.63 | 65.78 | 15.54 | 3.716 / 0.869 |
| 0.1 | 77.09 | 74.61 | 12.64 | 3.454 / 0.850 |
| **0 (RayDF)** | **86.34** | **84.23** | **7.97** | **3.388 / 0.663** |

**Analysis:** Table 5 (1) clearly shows that, without the aid of our dual-ray visibility classifier, the main ray-surface distance field completely collapses and cannot predict reasonable distance values for novel rays in the test set. In Table 5 (2) and (3), with the input of surface point position, the classifier achieves higher accuracy and higher F1-score, thus providing the ray-surface distance network with more accurate visibility information to predict precise distance values. In addition, concatenating two input rays directly disrupts the inherent symmetry of input rays. This is shown in Table 5 (4), where the classifier trained in this way achieves high accuracy but exhibits a low F1-score. This indicates that such a classifier is significantly less robust than the one trained with symmetric input rays. Table 6 demonstrates that a less robust classifier results in the ray-surface distance network predicting inaccurate distances.

In addition to the dual-ray visibility classifier, we also perform various ablation studies on our entire pipeline on the Blender dataset.

**Ablation on Multi-view Rays Sampling:** We conduct an ablation study with different values of $M \in \{10, 20, 40\}$ to determine the optimal number of multi-view rays for training a ray-surface distance network. Table 7 demonstrates that increasing the sampled multi-view rays enhances the accuracy of the ray-surface distance network. Although $M = 40$ achieves lower ADE and CD values, the substantial GPU memory and extended training time outweigh the limited performance improvement. Therefore, we opt to sample multi-view rays with $M = 20$ to train our ray-surface distance network in all main experiments.

Table 7: Ablation study on the number of multi-view rays sampling.

| $M$ | ADE↓ | CD ($\times 10^{-3}$)↓ mean / median |
|---|---|---|
| 10 | 10.12 | 4.469 / 0.713 |
| **20 (RayDF)** | 7.97 | 3.388 / 0.663 |
| 40 | **7.72** | **2.711 / 0.621** |

**Ablation on Sparse Depth Supervision:** With the advancement of existing techniques of depth estimation from RGB images, it is quite feasible to obtain sparse depth signals using existing techniques such as SfM or learning-based monocular depth estimators. In this regard, we additionally provide experimental results using sparse depth supervision. In Table 8, our method maintains satisfactory performance, even with only 1% depth values during training. We hypothesize that such robustness comes from our multi-view consistency constraint, because many depth values in the training set may be redundant thanks to our effective classifier.

Table 8: Ablation study on sparse depth supervision.

| Depth Sparsity | ADE↓ | CD ($\times 10^{-3}$)↓ mean / median |
|---|---|---|
| 1% | 8.54 | **3.301** / 0.937 |
| 5% | 8.70 | 3.250 / 0.920 |
| 10% | 8.68 | 3.313 / 0.924 |
| **100% (RayDF)** | **7.97** | 3.388 / **0.663** |

**Ablation of One-stage RayDF:** In the paper, we adopt a two-stage training strategy for our dual-ray visibility classifier and ray-surface distance network. For a comparison, we simply train both networks simultaneously. However, not surprisingly, the performance of one-stage training drops noticeably as shown in Table 9. This drop in performance is primarily because the classifier is inaccurate at the early stage and unlikely to provide effective constraints for the ray-surface distance network, given the similar number of training steps. Nevertheless, exploring one-stage training remains an intriguing direction for our future research efforts.

Table 9: Ablation study on one/two-stage training.

| | ADE↓ | CD ($\times 10^{-3}$)↓ mean / median |
|---|---|---|
| one-stage | 12.41 | 4.032 / **0.659** |
| **two-stage (RayDF)** | **7.97** | **3.388** / 0.663 |

For a more comprehensive analysis, we provide extensive ablation studies and additional qualitative as well as detailed quantitative results of each scene in the Blender dataset in Appendix A.4.

## 5 Conclusion

In this paper, we have shown that it is truly possible to efficiently and accurately learn 3D shape representations by using a multi-view consistent ray-based framework. In contrast to the existing coordinate-based methods, we instead use a simple ray-surface distance field to represent 3D shape geometries, which is further driven by a novel dual-ray visibility classifier to be multi-view shape consistent. Extensive experiments on multiple datasets demonstrate the extremely high rendering efficiency and outstanding performance of our approach. It would be interesting to extend our framework with more advanced techniques such as faster implementation and additional regularizations.

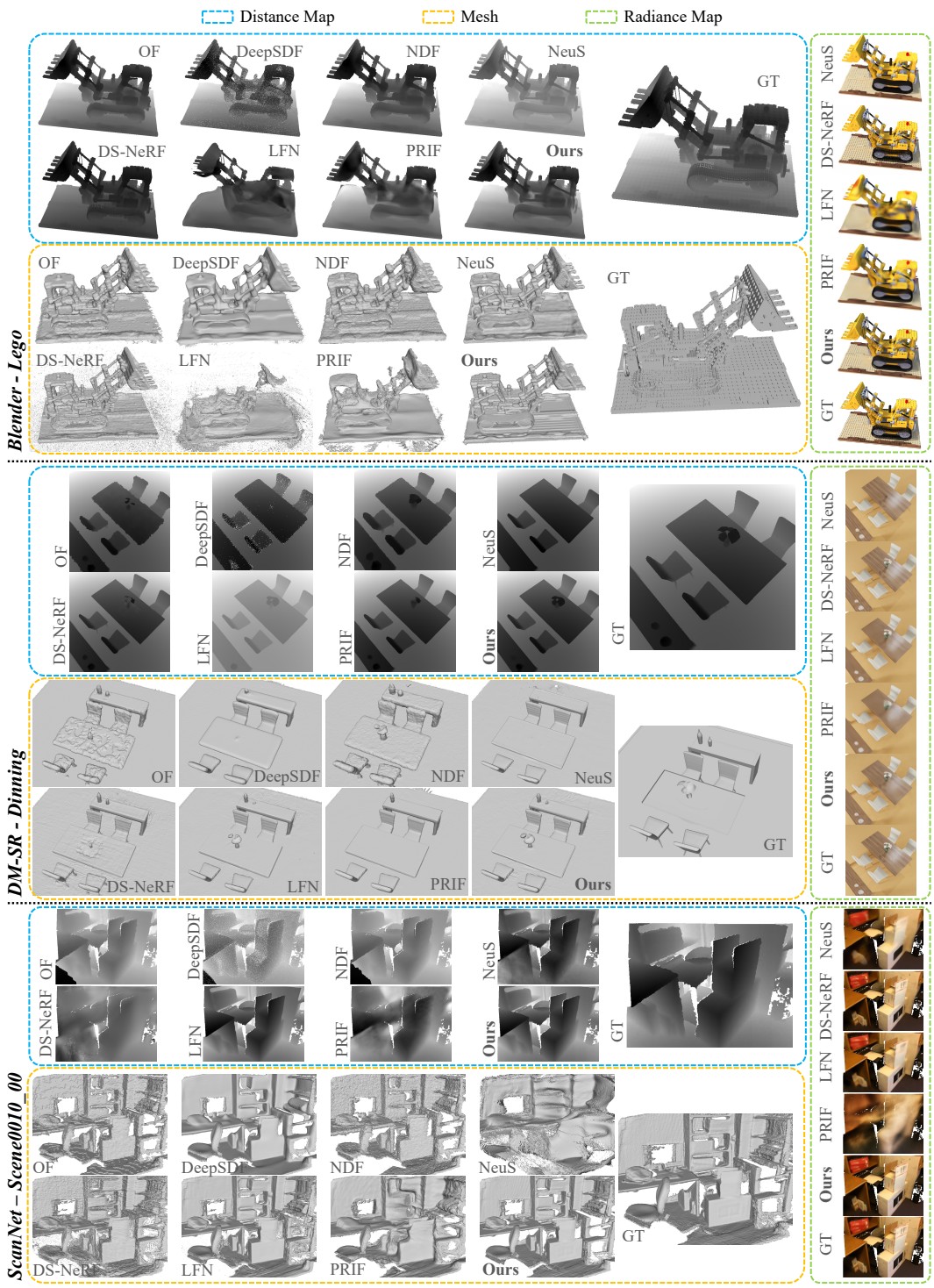

Figure 5: Qualitative results of all methods on the three datasets. More qualitative results can be found in Appendix A.7 and our project page: https://vlar-group.github.io/RayDF.html

**Acknowledgement:** This work was supported in part by Research Grants Council of Hong Kong under Grants 15225522 & 25207822 & 15606321, in part by National Natural Science Foundation of China under Grant 62271431, and in part by The Hong Kong Polytechnic University under Grant P0031501.

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

# A Appendix

## A.1 Network Architecture and Training Details

We provide the details of our ray-surface distance field and the auxiliary dual-ray visibility classifier.

### A.1.1 Ray-surface Distance Field

We use a 13-layer SIREN [63] with 1024 hidden units to learn a ray-surface distance field. We set the batch size as 8192 and train the network for 10 epochs with the Adam [35] optimizer and a cosine annealing strategy [44] with the learning rate initialized as $10^{-5}$ and decayed to $10^{-8}$.

**Radiance Branch:** To learn a radiance field, we take an additional 2-layer SIREN with 1024 hidden units as the radiance branch and output RGB values with sigmoid activation. For optimization together with the distance field, we adopt the same multi-view consistency as the distance field with a mean-squared loss and set the loss weight to 1.

### A.1.2 Dual-ray Visibility Classifier

We take a SIREN layer as the ray-feature encoder $g()$ and another SIREN layer as the point-feature encoder $k()$, followed by a 7-layer SIREN with 512 hidden units and output the visibility with sigmoid activation. Both the ray-feature encoder and the point-feature encoder are with 512 hidden units. The batch sizes are 2048 for Blender dataset and ScanNet dataset, and 1024 for DM-SR dataset. We train the network for 5 epochs with the Adam optimizer and a cycle annealing strategy [66] with the maximum learning rate $10^{-4}$.

### A.1.3 Training Data Generation

- For a scan with extrinsic parameters [R|t] and intrinsic parameters K, all rays of this scan start from the camera position $\boldsymbol{o} = (t_x, t_y, t_z)$. The ray orientation of pixel $(u, v)$ can be denoted as $\boldsymbol{m}_0 = \text{RK}^{-1}\left(\begin{smallmatrix} u \\ v \\ 1 \end{smallmatrix}\right)$, and the unit ray direction is $\boldsymbol{m} = \boldsymbol{m}_0 / \|\boldsymbol{m}_0\|$. Given an oriented ray $\boldsymbol{r}$ starting from $\boldsymbol{o}$ and a pre-defined sphere with a diameter $D$, we follow the two-sphere parameterization [8] to compute two intersections $\boldsymbol{p}^{in}, \boldsymbol{p}^{out}$ and construct our input parameters $\boldsymbol{r} = (\theta^{in}, \phi^{in}, \theta^{out}, \phi^{out})$.

- To obtain the ray-surface distance $d$ from the raw depth value $\dot{d}$ of pixel $(u, v)$, we have $d = \dot{d}\sqrt{(u - c_y)^2 + (v - c_x)^2 + f^2}/f - d_0$, where $f, c_y, c_x$ are the camera intrinsic parameters (*i.e.*, focal length, the center coordinates at y-axis and x-axis of image plane), and $d_0 = \|\boldsymbol{p}^{in} - \boldsymbol{o}\|$.

- For a ray $\boldsymbol{r}$ at the pixel $(u, v)$ of a specific scan, we have the first ray-sphere intersection $\boldsymbol{p}^{in}$ and ray direction $\boldsymbol{m}$, then we can compute the ray-surface point $\boldsymbol{p} = \boldsymbol{p}^{in} + d\boldsymbol{m}$. To reproject this point to the remaining $(K - 1)$ scans, for example, we calculate the pixel coordinate $(u^k, v^k)$ of the $k^{th}$ scan using its extrinsic parameters $[\text{R|t}]^k$ and intrinsic parameters $K^k$:

$$(u^k, v^k, 1) = \text{K}^k([\text{R|t}]^k)^{-1}\boldsymbol{p}/z_{\boldsymbol{p}} \tag{5}$$

  Then we can query the raw depth value at $(u^k, v^k)$ from the $k^{th}$ scan, and obtain its ray-surface distance $d^k_{u^k, v^k}$ and the ray parameters $\boldsymbol{r}^k_{u^k, v^k}$ (simplified as $d^k, \boldsymbol{r}^k$) following the previous steps. Besides, we calculate the multi-view distance $\tilde{d}^k$ between the ray-surface point $\boldsymbol{p}$ and the first intersection $\boldsymbol{p}^{in}_k$ of ray $\boldsymbol{r}^k$: $\tilde{d}^k = \|\boldsymbol{p} - \boldsymbol{p}^{in}_k\|$. We set a *closeness* threshold $\varepsilon = 10$ millimeters to determine whether the projected $k^{th}$ ray is visible:

$$v^k = \begin{cases} 1, & \text{if } |\tilde{d}^k - d^k| \leq \varepsilon \\ 0, & \text{otherwise} \end{cases} \tag{6}$$

- For both training and inference, the values of input parameters are normalized to be [-1, 1]. In particular, the latitude $\theta := 2\theta/\pi - 1$ and the longitude $\phi := \phi/\pi$. The input surface hitting point $(x_1, y_1, z_1)$ of our visibility classifier is also normalized to [-1, 1] by subtracting the sphere center and then dividing by the sphere radius $D/2$. The ground-truth ray-surface distance $d$ and the multi-view distance $\tilde{d}^k$ are scaled to be [0, 1] by dividing by the sphere diameter $D$.

## A.2 Derivation of Surface Normal

In this section, we derive the formula of surface normal from our ray-surface distance field.

As mentioned in Section A.1.3, given a ray starting from $\boldsymbol{o}$ with direction $\boldsymbol{m}$ and a fix-sized sphere with a diameter $D$ centered at the coordinate origin, we can construct the input parameters $\boldsymbol{r} = (\theta^{in}, \phi^{in}, \theta^{out}, \phi^{out})$ from a *ray-sphere intersect function* $F(\boldsymbol{o}, \boldsymbol{m}, D) = \boldsymbol{r}$.

By converting $\boldsymbol{m}$ to the spherical coordinate system, we have $\frac{\partial \boldsymbol{r}}{\partial \theta_{\boldsymbol{m}}}$ and $\frac{\partial \boldsymbol{r}}{\partial \phi_{\boldsymbol{m}}}$, which is the gradient of the above intersection function $F()$ *w.r.t.* the ray direction $\boldsymbol{m}$.

As shown in Figure 6, to compute the surface normal of a predicted surface point $\hat{\boldsymbol{p}}$ along the ray direction $\boldsymbol{m}$, we define a *sphere* of radius $R = D\hat{d} + d_0$ centered at the camera position $\boldsymbol{o}$ using spherical coordinates:

$$\Phi(\theta_{\boldsymbol{m}}, \phi_{\boldsymbol{m}}) = (R\sin\theta_{\boldsymbol{m}}\cos\phi_{\boldsymbol{m}}, R\sin\theta_{\boldsymbol{m}}\sin\phi_{\boldsymbol{m}}, R\cos\theta_{\boldsymbol{m}}) \tag{7}$$

In general, the formula for a unit normal vector is $\boldsymbol{n} = \frac{\frac{\partial \Phi}{\partial \phi_{\boldsymbol{m}}} \times \frac{\partial \Phi}{\partial \theta_{\boldsymbol{m}}}}{\|\frac{\partial \Phi}{\partial \phi_{\boldsymbol{m}}} \times \frac{\partial \Phi}{\partial \theta_{\boldsymbol{m}}}\|}$. From Eq. 7, we have

$$\frac{\partial \Phi}{\partial \phi_{\boldsymbol{m}}} = D \begin{bmatrix} (\frac{\partial \hat{d}}{\partial \phi_{\boldsymbol{m}}}\cos\phi_{\boldsymbol{m}} - (\hat{d} + \frac{d_0}{D})\sin\phi_{\boldsymbol{m}})\sin\theta_{\boldsymbol{m}} \\ (\frac{\partial \hat{d}}{\partial \phi_{\boldsymbol{m}}}\sin\phi_{\boldsymbol{m}} + (\hat{d} + \frac{d_0}{D})\cos\phi_{\boldsymbol{m}})\sin\theta_{\boldsymbol{m}} \\ \frac{\partial \hat{d}}{\partial \phi_{\boldsymbol{m}}}\cos\theta_{\boldsymbol{m}} \end{bmatrix} \tag{8}$$

$$\frac{\partial \Phi}{\partial \theta_{\boldsymbol{m}}} = D \begin{bmatrix} (\frac{\partial \hat{d}}{\partial \theta_{\boldsymbol{m}}}\sin\theta_{\boldsymbol{m}} + (\hat{d} + \frac{d_0}{D})\cos\theta_{\boldsymbol{m}})\cos\phi_{\boldsymbol{m}} \\ (\frac{\partial \hat{d}}{\partial \theta_{\boldsymbol{m}}}\sin\theta_{\boldsymbol{m}} + (\hat{d} + \frac{d_0}{D})\cos\theta_{\boldsymbol{m}})\sin\phi_{\boldsymbol{m}} \\ \frac{\partial \hat{d}}{\partial \theta_{\boldsymbol{m}}}\cos\theta_{\boldsymbol{m}} - (\hat{d} + \frac{d_0}{D})\sin\theta_{\boldsymbol{m}} \end{bmatrix} \tag{9}$$

Here, we can replace $\frac{\partial \hat{d}}{\partial \theta_{\boldsymbol{m}}} = \frac{\partial \hat{d}}{\partial \boldsymbol{r}}\frac{\partial \boldsymbol{r}}{\partial \theta_{\boldsymbol{m}}}$ and $\frac{\partial \hat{d}}{\partial \phi_{\boldsymbol{m}}} = \frac{\partial \hat{d}}{\partial \boldsymbol{r}}\frac{\partial \boldsymbol{r}}{\partial \phi_{\boldsymbol{m}}}$, so the surface normal $\boldsymbol{n}$ can be denoted as a function $Q(\frac{\partial \hat{d}}{\partial \boldsymbol{r}}, \boldsymbol{r}, D, \hat{d})$.

## A.3 Additional Implementation Details

In this section, we provide details of implementation and some minor adaptations of baselines, as well as the workflow for 3D shape reconstruction.

### A.3.1 Baselines

**OF/DeepSDF/NDF:** Pre-processing on mesh before training is required for OF/DeepSDF/NDF. For fair comparisons, we use all depth images from training views to reconstruct the training mesh, which is then used for sampling occupancies/signed distances/unsigned distances of a $256^3$ voxel grid for training. We follow the official settings of hyperparameters and network architecture to train on the three datasets. For inference, we apply

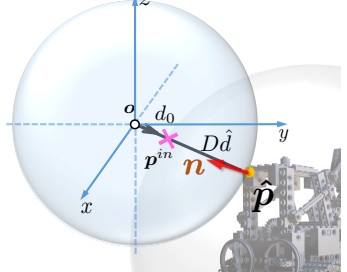

Figure 6: Sphere for surface normal computation.

sphere tracing for DeepSDF and NDF to render depth images from testing views. The sphere tracing will stop after 100 iterations or when the predicted distance is smaller than a threshold at 0.005 meters. To avoid missing the intersections, we follow the damped sphere tracing [14] from NDF, *i.e.*, marching with $\alpha \cdot f(\boldsymbol{p})$ with $\alpha = 0.6$. Since sphere tracing for OF can only take a fixed step size (*e.g.*, 0.005 meters), it requires about 1000 iterations to march the surface, which is time-consuming. We thus construct a testing mesh by querying a value grid of occupancies and use ray-mesh intersections to render depth images for evaluation.

**NeuS/DS-NeRF:** In general, we follow the same network architecture, training schedule, and color supervision from NeuS and DS-NeRF. The vanilla DS-NeRF adopts sparse 3D points from structure-from-motion (SFM) and uses the reprojection error for depth supervision. For a fair comparison, we provide the same dense depth supervision from training views as ours on all three datasets for both

NeuS and DS-NeRF. For the experiments of Group 1, *i.e.*, 3D shape representation only, we remove the color supervision of NeuS and DS-NeRF and leave depth supervision only.

**LFN/PRIF:** We adopt the same network architecture and hyperparameters from the official settings of LFN/PRIF. In particular, we train LFN/PRIF for 10 epochs on each dataset, which is the same as ours. Since the proposed depth computation from LFN makes it difficult to obtain dense depths, we modify the output of the last MLP layer to predict depth value with supervision. For the experiments of Group 2, *i.e.*, 3D Shape and Appearance Representations, we extend a 2-layer color branch both for LFN and PRIF with appearance supervision. Note that, the LFN/PRIF results are obtained without our dual-ray visibility classifier. We additionally conduct experiments on LFN/PRIF with our own classifier for comparison. Figure 7 and Table 10 show that our method still achieves the best performance.

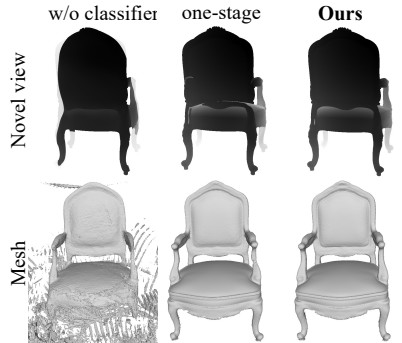

Figure 7: Qualitative results of LFN/PRIF with our visibility classifier.

### A.3.2 3D Shape Reconstruction for Evaluation

For a fair comparison, we initially render depth images of testing views for all baselines and ours. For OF/DeepSDF/NDF, we set the depth value to 0 for those unconverged rays (*i.e.*, no intersection during sphere tracing). For PRIF and ours, we apply specific outlier removal strategies. Subsequently, we use TSDF fusion on the processed depth images to obtain complete meshes for further evaluation.

Table 10: Quantitative results of LFN/PRIF with our visibility classifier.

| | ADE↓ | CD ($\times 10^{-3}$)↓ mean / median |
|---|---|---|
| LFN + Our Visibility Classifier | 90.82 | 121.289 / 44.564 |
| PRIF + Our Visibility Classifier | 9.97 | 4.187 / 0.891 |
| **RayDF** | **7.97** | **3.388 / 0.663** |

## A.4 Additional Ablation Studies

In this section, we show more details of ablation studies, including ablations on dual-ray visibility classifier, ray-surface distance network, training strategies, and post-processing.

### A.4.1 Ablation Studies on the Dual-ray Visibility Classifier

The key issue of a naïve ray based distance function is the lack of generalization ability across novel views during testing, fundamentally because training a single ray based network (without classifier) tends to fit all ray distance data pairs of the training set, but cannot guarantee the consistency of surface distances between seen (training) rays and unseen (testing) rays. As demonstrated in Tables 11, 12, 13 and Figure 8, the absence of the classifier significantly impacts testing performance, resulting in notably inferior results compared to those optimized with the well-trained classifier. To provide better clarity, we also present visualizations of the visibility predictions in Figure 9. Moreover, the results of our classifier with noises in Tables 14, 15, 16 and Figure 10 demonstrate that our visibility classifier is reliable for multi-view consistency.

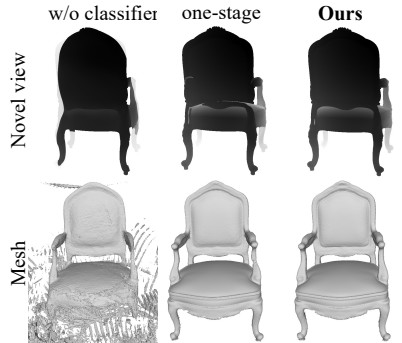
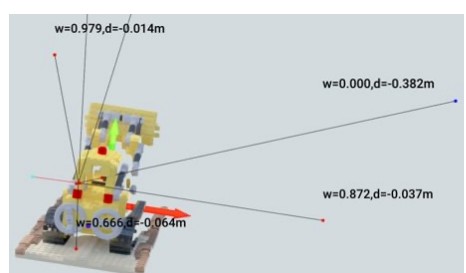

Figure 8: Qualitative results of RayDF without visibility classifier and one-stage trained RayDF.

Figure 9: Visualization of visibility predictions, *w* denotes visibility, *d* denotes distance difference.

Table 11: Ablation study on the visibility classifier. Accuracy (%)↑ and F1-Score (%)↑ on 8 scenes and the average scores of the Blender dataset.

| | Chair Acc. | Chair F1 | Drums Acc. | Drums F1 | Ficus Acc. | Ficus F1 | Hotdog Acc. | Hotdog F1 | Lego Acc. | Lego F1 | Materials Acc. | Materials F1 | Mic Acc. | Mic F1 | Ship Acc. | Ship F1 | Avg. Acc. | Avg. F1 |
|---|---|---|---|---|---|---|---|---|---|---|---|---|---|---|---|---|---|---|
| (1) w/o classifier $h_\Phi$ | - | - | - | - | - | - | - | - | - | - | - | - | - | - | - | - | - | - |
| (2) use $d$ in function $k$ | 91.72 | 91.82 | 82.45 | 81.87 | 65.14 | 60.00 | 93.11 | 93.73 | 80.06 | 74.98 | 75.62 | 51.96 | 81.53 | 81.92 | 82.81 | 85.06 | 81.56 | 77.67 |
| (3) remove function $k$ | 91.37 | 91.55 | 80.06 | 78.51 | 64.10 | 48.07 | 93.33 | 94.44 | 81.55 | 75.86 | 80.55 | 60.55 | 82.77 | 82.88 | 81.74 | 82.32 | 81.93 | 76.77 |
| (4) use $g(\boldsymbol{r}_1)\oplus g(\boldsymbol{r}_2)$ | **97.56** | 64.57 | **93.96** | 54.01 | **95.53** | 23.75 | **96.31** | 51.63 | **93.82** | 52.37 | **96.04** | 31.77 | **97.56** | 64.04 | **89.67** | 59.24 | **95.06** | 50.17 |
| **Dual-ray Visibility Classifier** | 92.92 | **93.22** | 84.98 | **84.73** | 70.34 | **61.83** | 93.61 | 94.67 | 87.74 | **85.24** | 89.95 | **81.08** | 87.00 | **87.78** | 84.16 | **85.26** | 86.34 | **84.23** |

Table 12: Ablation study on the visibility classifier. ADE↓ on 8 scenes and the average score of the Blender dataset.

| | Chair | Drums | Ficus | Hotdog | Lego | Materials | Mic | Ship | Avg. |
|---|---|---|---|---|---|---|---|---|---|
| (1) w/o classifier $h_\Phi$ | 30.39 | 66.12 | 54.93 | 23.92 | 32.19 | 84.10 | 52.54 | 32.46 | 47.08 |
| (2) use $d$ in function $k$ | 6.14 | 12.57 | 15.43 | 4.15 | 13.44 | 12.49 | 9.19 | 13.15 | 10.82 |
| (3) remove function $k$ | 6.24 | 11.57 | 15.39 | 4.08 | 12.08 | 10.42 | 8.64 | 13.45 | 10.23 |
| (4) use $g(\boldsymbol{r}_1)\oplus g(\boldsymbol{r}_2)$ | 5.89 | 11.69 | 14.29 | 4.70 | 11.01 | 7.91 | 8.84 | 13.85 | 9.77 |
| **Full RayDF** | **4.59** | **9.24** | **12.56** | **3.06** | **7.98** | **5.68** | **7.55** | **13.07** | **7.97** |

Table 13: Ablation study on the visibility classifier. CD ($\times 10^{-3}$)↓ (mean / median) on 8 scenes and the average score of the Blender dataset.

| | Chair | Drums | Ficus | Hotdog | Lego | Materials | Mic | Ship | Avg. |
|---|---|---|---|---|---|---|---|---|---|
| (1) w/o classifier $h_\Phi$ | 892.472 / 856.377 | 1170.313 / 1064.195 | 2077.365 / 2038.962 | 1052.000 / 541.289 | 1088.382 / 822.446 | 1879.326 / 1565.162 | 1394.594 / 1277.556 | 1077.019 / 423.644 | 1328.934 / 1073.704 |
| (2) use $d$ in function $k$ | **3.286** / 0.426 | 6.368 / 2.021 | 7.390 / 0.789 | **7.020** / 0.768 | 3.114 / 0.858 | 3.618 / 1.118 | 3.730 / 1.110 | 5.349 / 0.863 | 4.984 / 0.994 |
| (3) remove function $k$ | 4.312 / 0.437 | 5.480 / 1.637 | 7.497 / 0.588 | 7.527 / 0.778 | 3.010 / 0.848 | 2.967 / 0.956 | 2.613 / 1.035 | 4.614 / 0.837 | 4.753 / 0.890 |
| (4) use $g(\boldsymbol{r}_1)\oplus g(\boldsymbol{r}_2)$ | 7.362 / 0.424 | 4.930 / 1.513 | 5.473 / 0.647 | 7.111 / 0.782 | 1.719 / 0.745 | 1.832 / 0.793 | **2.417** / 0.829 | 7.078 / 0.866 | 4.740 / 0.825 |
| **Full RayDF** | 3.807 / **0.312** | 3.310 / 0.958 | 2.914 / 0.543 | 7.255 / **0.733** | **1.095** / 0.702 | **1.604** / 0.617 | 3.571 / **0.644** | **3.544** / 0.792 | **3.388** / **0.663** |

Table 14: Ablation study on adding noises to the well-trained visibility classifier. Accuracy (%)↑ and F1-Score (%)↑ on 8 scenes and the average scores of the Blender dataset.

| Noise level ($\sigma^2$) | Chair Acc. | Chair F1 | Drums Acc. | Drums F1 | Ficus Acc. | Ficus F1 | Hotdog Acc. | Hotdog F1 | Lego Acc. | Lego F1 | Materials Acc. | Materials F1 | Mic Acc. | Mic F1 | Ship Acc. | Ship F1 | Avg. Acc. | Avg. F1 |
|---|---|---|---|---|---|---|---|---|---|---|---|---|---|---|---|---|---|---|
| 1.0 | 65.04 | 66.57 | 61.75 | 63.31 | 55.19 | 51.58 | 65.91 | 70.55 | 62.56 | 59.06 | 63.79 | 50.11 | 62.92 | 65.24 | 62.92 | 65.24 | 62.51 | 61.46 |
| 0.5 | 70.46 | 71.61 | 65.99 | 66.82 | 57.04 | 51.57 | 71.64 | 75.67 | 63.99 | 67.76 | 68.75 | 53.99 | 67.59 | 69.42 | 67.59 | 69.42 | 66.63 | 65.78 |
| 0.1 | 85.02 | 85.37 | 77.31 | 76.32 | 61.78 | 49.78 | 86.96 | 88.96 | 68.01 | 72.59 | 81.98 | 66.75 | 80.23 | 81.01 | 75.44 | 76.08 | 77.09 | 74.61 |
| **0 (RayDF)** | **92.92** | **93.22** | **84.98** | **84.73** | **70.34** | **61.83** | **93.61** | **94.67** | **87.74** | **85.24** | **89.95** | **81.08** | **87.00** | **87.78** | **84.16** | **85.26** | **86.34** | **84.23** |

Table 15: Ablation study on adding noises to the well-trained visibility classifier. ADE↓ on 8 scenes and the average score of the Blender dataset.

| Noise level ($\sigma^2$) | Chair | Drums | Ficus | Hotdog | Lego | Materials | Mic | Ship | Avg. |
|---|---|---|---|---|---|---|---|---|---|
| 1.0 | 10.31 | 17.08 | 17.07 | 18.21 | 23.09 | 19.02 | 11.33 | 20.09 | 17.03 |
| 0.5 | 9.11 | 15.67 | 16.28 | 16.33 | 21.16 | 15.92 | 10.60 | 19.24 | 15.54 |
| 0.1 | 7.05 | 12.91 | 15.04 | 11.92 | 16.83 | 10.51 | 9.46 | 17.39 | 12.64 |
| **0 (RayDF)** | **4.59** | **9.24** | **12.56** | **3.06** | **7.98** | **5.68** | **7.55** | **13.07** | **7.97** |

Table 16: Ablation study on adding noises to the well-trained visibility classifier. CD ($\times 10^{-3}$)↓ (mean / median) on 8 scenes and the average score of the Blender dataset.

| Noise level ($\sigma^2$) | Chair | Drums | Ficus | Hotdog | Lego | Materials | Mic | Ship | Avg. |
|---|---|---|---|---|---|---|---|---|---|
| 1.0 | 3.954 / 0.329 | 3.353 / 1.299 | 3.068 / 0.630 | 7.474 / 0.963 | 2.844 / 0.705 | 3.310 / 1.045 | 2.841 / 1.564 | 5.415 / 0.943 | 4.032 / 0.935 |
| 0.5 | 3.815 / 0.287 | **3.187** / 1.221 | 2.793 / **0.533** | 7.080 / 0.933 | 2.313 / 0.658 | 2.923 / 0.959 | **2.817** / 1.429 | 4.796 / 0.933 | 3.716 / 0.869 |
| 0.1 | 3.808 / **0.276** | 3.483 / 1.103 | **2.395** / 0.548 | **6.730** / 0.857 | 1.739 / **0.622** | 2.840 / 1.012 | 2.883 / 1.510 | 3.751 / 0.875 | 3.454 / 0.850 |
| **0 (RayDF)** | **3.807** / 0.312 | 3.310 / **0.958** | 2.914 / 0.543 | 7.255 / **0.733** | **1.095** / 0.702 | **1.604** / **0.617** | 3.571 / **0.644** | **3.544** / **0.792** | **3.388** / **0.663** |

### A.4.2 Ablation Studies on the Ray-surface Distance Network

We perform the following ablation studies on the ray-surface distance network.

**(1) Multi-view Rays Sampling:** Here we report the detailed quantitative results of each scene on Blender dataset in Tables 17 and 18.

**(2) Network Architecture:** We conduct additional experiments on the Blender dataset as shown in Table 19. We can see that our framework tends to favor a wider or deeper network. However, to extensively explore an optimal network architecture is non-trivial and we leave it for our future work.

**(3) Multi-view Consistency Loss $\ell_{mv}$:** In our multi-view consistency loss function $\ell_{mv}$(Eq. 3), we simply change it from $\ell_1$ to $\ell_2$ optimization, and conduct an additional experiment on *Reception* of DMSR dataset. From Table 20 and Figure 12, we can see that $\ell_2$ based loss function can better reconstruct thin structures but with more noisy results than $\ell_1$ based loss. In this regard, it is of great interest to fully address the recovering task of thin structures and we leave it for future exploration.

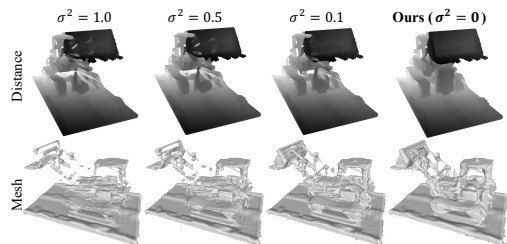

Figure 10: Qualitative results of RayDF using visibility classifier with noises.

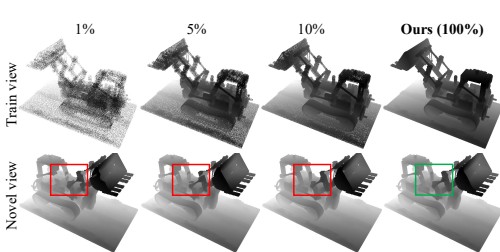

Figure 11: Qualitative results of using sparse depth values in supervision.

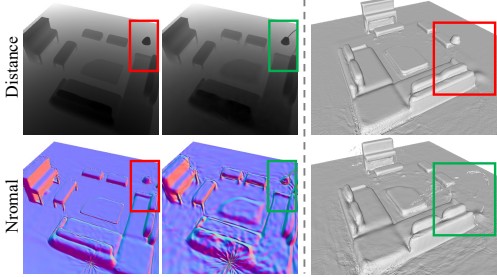

Figure 12: Qualitative results of RayDF on *Reception* in DM-SR dataset. *Red*: RayDF-$l_1$, *Green*: RayDF-$l_2$.

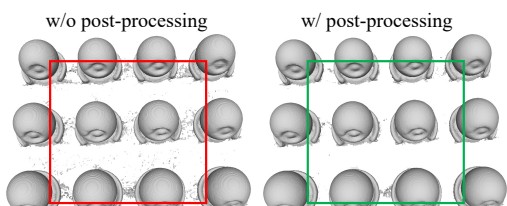

Figure 13: Qualitative results of the predicted mesh by RayDF with and without post-processing.

Table 17: Ablation study on the number of multi-view rays sampling. ADE↓ on 8 scenes and the average score of the Blender dataset.

| $M$ | Chair | Drums | Ficus | Hotdog | Lego | Materials | Mic | Ship | Avg. |
|---|---|---|---|---|---|---|---|---|---|
| 10 | 4.89 | 9.82 | 13.27 | 3.14 | 8.27 | 6.44 | 7.71 | 13.29 | 10.12 |
| **20 (RayDF)** | 4.59 | 9.24 | 12.56 | 3.06 | 7.98 | 5.68 | 7.55 | 13.07 | 7.97 |
| 40 | **4.46** | **8.84** | **12.05** | **2.90** | **7.78** | **5.37** | **7.48** | **12.91** | **7.72** |

Table 18: Ablation study on the number of multi-view rays sampling. CD $(\times 10^{-3})$↓ (mean / median) on 8 scenes and the average score of the Blender dataset.

| $M$ | Chair | Drums | Ficus | Hotdog | Lego | Materials | Mic | Ship | Avg. |
|---|---|---|---|---|---|---|---|---|---|
| 10 | 6.341 / 0.328 | 5.680 / 1.057 | 5.789 / 0.630 | **7.052 / 0.697** | 1.235 / 0.734 | 1.743 / 0.675 | 3.724 / 0.789 | 4.190 / 0.792 | 4.469 / 0.713 |
| **20 (RayDF)** | 3.807 / **0.312** | 3.310 / 0.958 | 2.914 / 0.543 | 7.255 / 0.733 | 1.095 / 0.702 | 1.604 / 0.617 | 3.571 / 0.644 | 3.544 / 0.792 | 3.388 / 0.663 |
| 40 | **3.445** / 0.314 | **2.395 / 0.865** | **2.616 / 0.508** | 7.264 / 0.742 | **1.080 / 0.672** | **0.918 / 0.531** | **1.508 / 0.579** | **2.463 / 0.758** | **2.711 / 0.621** |

Table 19: Ablation study on the network architecture of ray-surface distance network.

| | ADE↓ | CD $(\times 10^{-3})$ ↓ mean / median |
|---|---|---|
| 8 layers, 512 units | 9.13 | 4.311 / 0.954 |
| 8 layers, 1024 units | 8.52 | 4.010 / 0.805 |
| 13 layers, 512 units | 8.80 | 4.085 / 0.854 |
| **13 layers, 1024 units (RayDF)** | **7.97** | **3.388 / 0.663** |

Table 20: Ablation study on the multi-view consistency loss function on *Reception* in DM-SR dataset.

| | ADE↓ | CD $(\times 10^{-3})$ ↓ mean / median |
|---|---|---|
| $l_2$ | **6.56** | 12.141 / 6.853 |
| $l_1$ **(RayDF)** | 6.96 | **10.632 / 5.714** |

### A.4.3   Ablation Studies on Training Strategies

**(1) Sparse Depth Supervision:** We additionally provide experimental results of our method using sparse depth values in supervision in Tables 21, 22 and Figure 11.

**(2) One-stage RayDF:** We present detailed quantitative and qualitative results of one-stage training of our pipeline, the performance drops noticeably as shown in Tables 23, 24 and Figure 8.

### A.4.4   Ablation Studies on Ray Parameterization

**(1) Choices of Ray Parameterization:** We discuss our ray parameterization in detail in Section A.1.3. In fact, our pipeline supports various ray parameterizations. To verify this, we conduct ablation studies by replacing our spherical coordinate with the one used in LFN/PRIF. As shown in Table 25 and Figure 14, PRIF parameterization can also achieve satisfactory performance, while LFN parameterization is inferior in our pipeline.

Table 21: Ablation study on sparse depth supervision. ADE↓ on 8 scenes and the average score of the Blender dataset.

| Depth Sparsity | Chair | Drums | Ficus | Hotdog | Lego | Materials | Mic | Ship | Avg. |
|---|---|---|---|---|---|---|---|---|---|
| 1% | **4.29** | 11.06 | 15.06 | 3.21 | 8.69 | 4.84 | 8.33 | **12.87** | 8.54 |
| 5% | 4.83 | 10.63 | 14.44 | 3.65 | 8.55 | **4.58** | 8.44 | 14.48 | 8.70 |
| 10% | 4.85 | 10.15 | 14.36 | 3.99 | 8.06 | 4.78 | 8.22 | 15.01 | 8.68 |
| **100% (RayDF)** | 4.59 | **9.24** | **12.56** | **3.06** | **7.98** | 5.68 | **7.55** | 13.07 | **7.97** |

Table 22: Ablation study on sparse depth supervision. CD ($\times 10^{-3}$)↓ (mean / median) on 8 scenes and the average score of the Blender dataset.

| Depth Sparsity | Chair | Drums | Ficus | Hotdog | Lego | Materials | Mic | Ship | Avg. |
|---|---|---|---|---|---|---|---|---|---|
| 1% | 3.625 / 0.498 | 3.276 / 1.080 | 2.365 / 0.544 | 6.844 / 0.912 | 1.517 / 0.937 | 2.446 / 1.048 | **2.922** / 1.556 | **3.415** / 0.919 | **3.301** / 0.937 |
| 5% | 3.536 / 0.348 | **3.067** / 1.078 | 2.323 / 0.578 | **6.577** / 0.875 | 1.522 / 0.928 | 2.422 / 1.091 | 2.927 / 1.532 | 3.623 / 0.930 | 3.250 / 0.920 |
| 10% | **3.486** / 0.341 | 3.068 / 1.088 | **2.307** / 0.593 | 6.867 / 0.871 | 1.503 / 0.952 | 2.381 / 1.076 | **2.922** / 1.533 | 3.969 / 0.934 | 3.313 / 0.924 |
| **100% (RayDF)** | 3.807 / **0.312** | 3.310 / **0.958** | 2.914 / **0.543** | 7.255 / **0.733** | **1.095** / **0.702** | **1.604** / **0.617** | 3.571 / **0.644** | 3.544 / **0.792** | 3.388 / **0.663** |

Table 23: Ablation study on one/two-stage training. ADE↓ on 8 scenes and the average score of the Blender dataset.

| | Chair | Drums | Ficus | Hotdog | Lego | Materials | Mic | Ship | Avg. |
|---|---|---|---|---|---|---|---|---|---|
| one-stage | 6.69 | 12.35 | 17.37 | 10.94 | 15.88 | 9.20 | 9.27 | 17.61 | 12.41 |
| **two-stage (RayDF)** | **4.59** | **9.24** | **12.56** | **3.06** | **7.98** | **5.68** | **7.55** | **13.07** | **7.97** |

Table 24: Ablation study on one/two-stage training. CD ($\times 10^{-3}$)↓ (mean / median) on 8 scenes and the average score of the Blender dataset.

| | Chair | Drums | Ficus | Hotdog | Lego | Materials | Mic | Ship | Avg. |
|---|---|---|---|---|---|---|---|---|---|
| one-stage | 4.771 / **0.201** | 3.459 / 1.202 | 2.956 / **0.310** | 10.068 / **0.557** | 1.515 / **0.616** | 2.875 / 1.030 | **2.881** / 1.450 | 3.727 / **0.700** | 4.032 / **0.659** |
| **two-stage (RayDF)** | **3.807** / 0.312 | **3.310** / 0.958 | **2.914** / 0.543 | **7.255** / 0.733 | **1.095** / **0.702** | **1.604** / **0.617** | 3.571 / **0.644** | **3.544** / 0.792 | **3.388** / 0.663 |

Table 25: Ablation studies on ray parameterization on Blender dataset.

| | ADE↓ | CD ($\times 10^{-3}$) ↓ mean / median |
|---|---|---|
| LFN param. + (Ray-surface Distance Network + Visibility Classifier) | 89.53 | 82.102 / 10.298 |
| PRIF param. + (Ray-surface Distance Network + Visibility Classifier) | 8.42 | 3.409 / **0.621** |
| (LFN param. + Ray-surface Distance Network) + (Our param. + Visibility Classifier) | 91.30 | 142.570 / 74.345 |
| (PRIF param. + Ray-surface Distance Network) + (Our param. + Visibility Classifier) | 8.44 | 3.526 / 0.653 |
| **RayDF** | **7.97** | **3.388** / 0.663 |

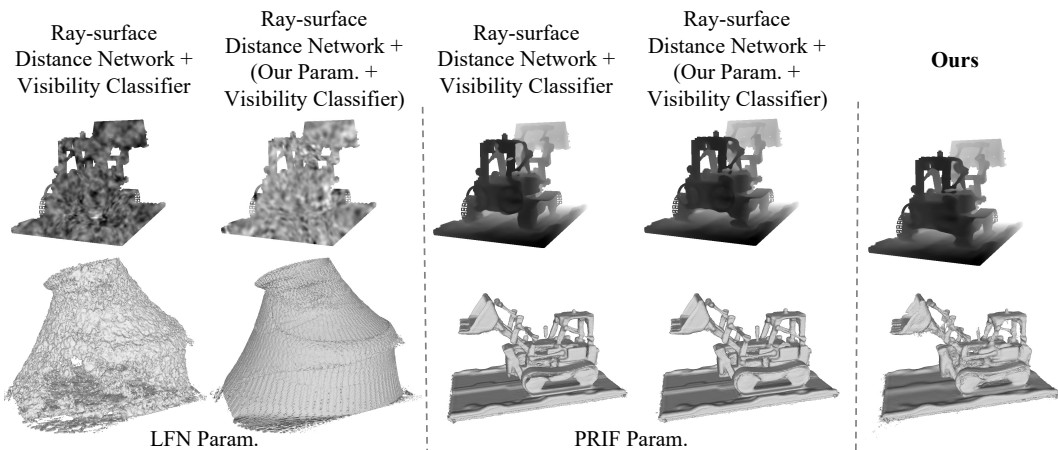

Figure 14: Qualitative results of using different ray parameterizations.

**(2) Artifacts from Spherical Coordinate:** The artifacts of fixed patterns in the predicted distance maps (as shown in Figure 15 (*red*)) are caused by the choice of spherical coordinate. We conduct additional experiments given two different choices of ray parameterizations. As shown in Table 25, our applied spherical coordinate can achieve the best ADE scores, and the caused artifacts can be easily fixed by using PRIF's ray parameterization as shown in Figure 15 (*green*). We leave it for our future work to fully address such an issue.

Table 26: Ablation study on post-processing. CD ($\times 10^{-3}$)↓ (mean / median) on 8 scenes and the average score of the Blender dataset.

| Post-processing | Chair | Drums | Ficus | Hotdog | Lego | Materials | Mic | Ship | Avg. |
|---|---|---|---|---|---|---|---|---|---|
| | 4.378 / **0.203** | 3.414 / 1.058 | 4.539 / **0.236** | 9.846 / **0.508** | 1.749 / 0.832 | 2.546 / 1.159 | **2.955** / 1.540 | 3.829 / **0.678** | 4.157 / 0.777 |
| ✓ | **3.807** / 0.312 | **3.310** / **0.958** | **2.914** / 0.543 | **7.255** / 0.733 | **1.095** / **0.702** | **1.604** / **0.617** | 3.571 / **0.644** | **3.544** / 0.792 | **3.388** / **0.663** |

### A.4.5  Ablation Studies on Post-processing

We follow the outlier removal step in Section 3.4 as post-processing to obtain clean 3D point clouds for further mesh reconstruction and evaluation. In all experiments, we set the outlier threshold as 5 for post-processing. We conduct additional experiments w/ and w/o the outlier removal step on the Blender dataset. In Table 26, the experiment without post-processing yields higher CD errors compared to the one with post-processing. The qualitative results are shown in Figure 13.



Figure 15: Visualizations of distance prediction using different ray parameterization.

### A.5  Limitations of RayDF

Since our RayDF takes the conventional spherical parameterization for the input ray, *i.e.*, a sphere is predefined to bound the target 3D scene, our RayDF with such a parameterization cannot represent the 3D scene with rays starting from a position inside the sphere. However, the proposed distance field with a visibility classifier of our RayDF is flexible to different ray parameterizations. How to integrate distinct parameterizations of input rays for different types of 3D scenes with our ray-surface distance field and the dual-ray visibility classifier can be a future exploration direction.

### A.6  Additional Quantitative Results

In this section, we report the detailed quantitative results of each scene on the three datasets.

**Blender:** In Tables 28 and 29, we show the quantitative results of the experiments in Group 1. Our RayDF achieves significantly better performance than the other baselines on most scenes, especially on complex scenes (*e.g.*, Drums, Ficus, and Ship). In Tables 30, 31, 32, 33 and 34, we report the geometry and appearance metrics in Group 2. Our method achieves comparable performance with DS-NeRF for novel view synthesis while recovering the 3D shape with fewer outliers.

**DM-SR:** In Tables 35 and 36, we report the quantitative comparisons in Group 1 and our RayDF performs better than the other baselines on those indoor scenes with more complex objects. As shown in Tables 37, 38, 39, 40 and 41, our method still achieves high-quality results for novel view synthesis on all scenes in Group 2 experiments.

**ScanNet:** In Tables 42 and 43, our RayDF performs significantly better than the other baselines on the real-world scenes. To represent these real-world 3D scenes with much more noise and complex geometry, our method outperforms all baselines while obtaining photo-realistic novel view synthesis results, as shown in Tables 44, 45, 46, 47 and 48.

Moreover, we provide comparisons of optimization time with baselines in Table 27. We compare the average training time of our method and all baselines on each scene of Blender dataset, using a single NVIDIA RTX 3090 GPU card and a CPU of AMD Ryzen 7. Since we apply a two-stage training strategy, our method is not as fast as baselines during optimization. Nevertheless, once our ray-surface distance network is optimized, it can achieve superior efficiency in rendering novel views, as shown in Table 1.

Table 27: Optimization time consumption (hours).

| | Time |
|---|---|
| OF [46] | **0.2** |
| DeepSDF [54] | 0.6 |
| NDF [14] | 2.1 |
| NeuS [77] | 10.2 |
| DS-NeRF [19] | 22.8 |
| LFN [64] | 0.9 |
| PRIF [23] | 2.2 |
| **RayDF (Ours)** | 24.9 |

### A.7  Additional Qualitative Results

We provide additional qualitative results on the three datasets as follows, including distance map, mesh from Group 1, and radiance map, textured mesh from Group 2. We also report the visualizations of our surface normal, which is computed following Section A.2.

**Blender:** In Figures 16 and 17, we show the qualitative results on *Materials* and *Drums*. In particular, *Materials* is a scene that contains empty spaces between multiple objects. Our RayDF predicts

distance with fewer outliers near the object surfaces. Our surface normal indicates that our method recovers the 3D scene with a smoother surface while preserving more details of the complex geometry.

**DM-SR:** In Figures 18 and 19, we report the qualitative results on *Reception* and *Study*. Our RayDF predicts accurate distances and hence is capable of reconstructing more detailed structures.

**ScanNet:** In Figures 20 and 21, we show the qualitative results on *Scene0005_00* and *Scene0030_00*, which are challenging real-world scenes. Our method recovers the 3D real-world scenes with reasonable details while predicting realistic radiance results.

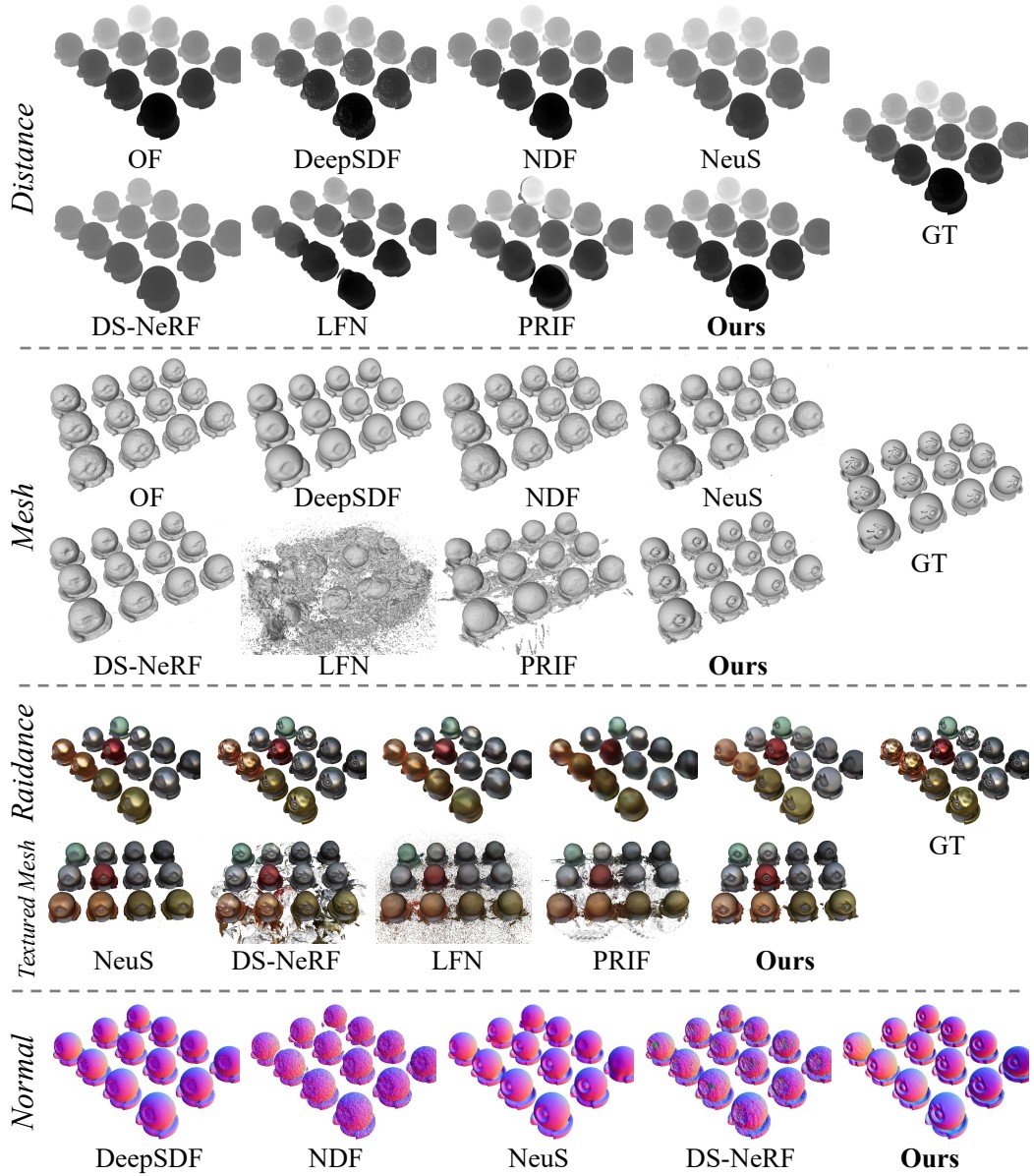

Figure 16: Qualitative results of all methods on *Materials* in Blender dataset.

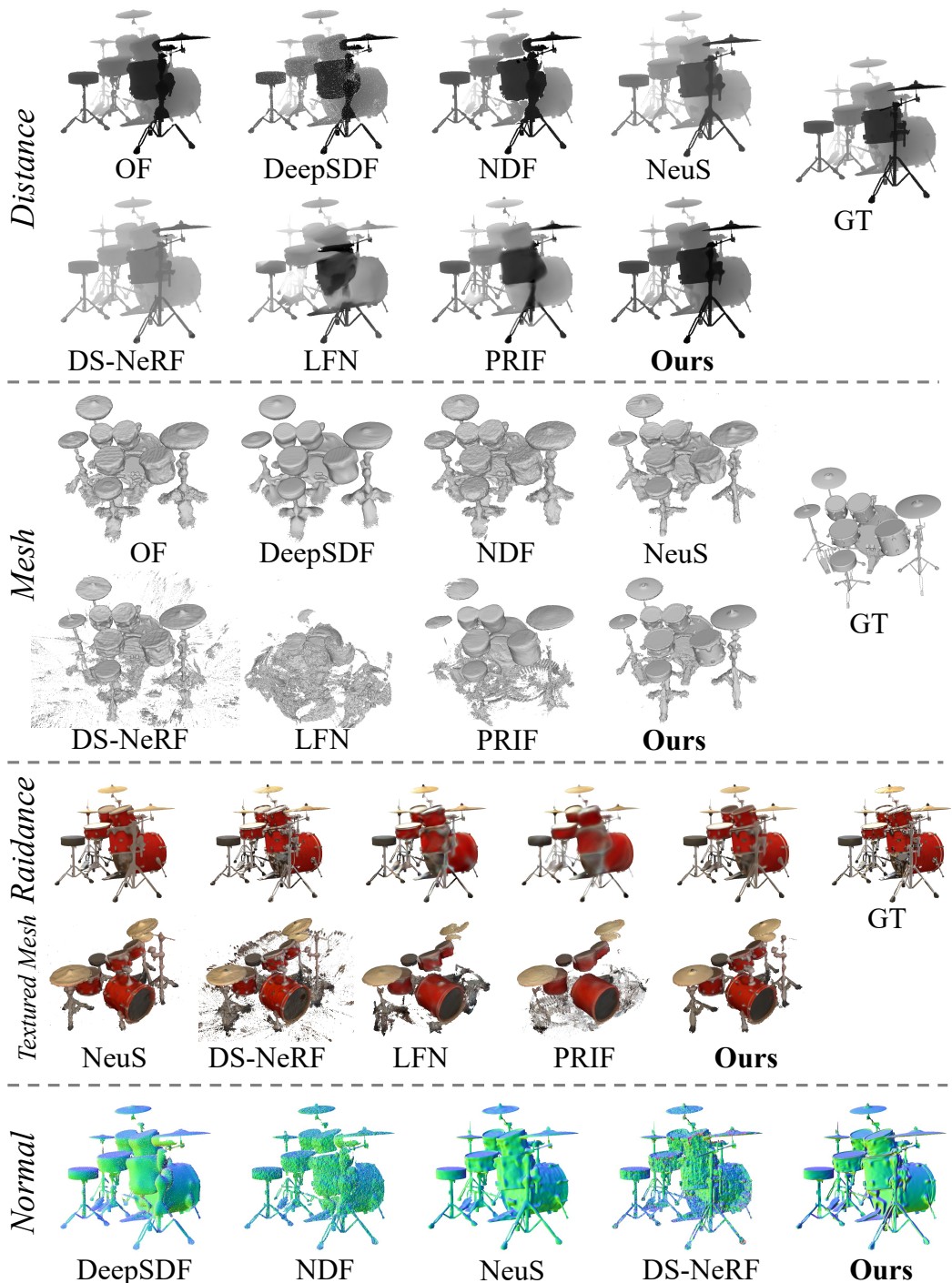

Figure 17: Qualitative results of all methods on *Drums* in Blender dataset.

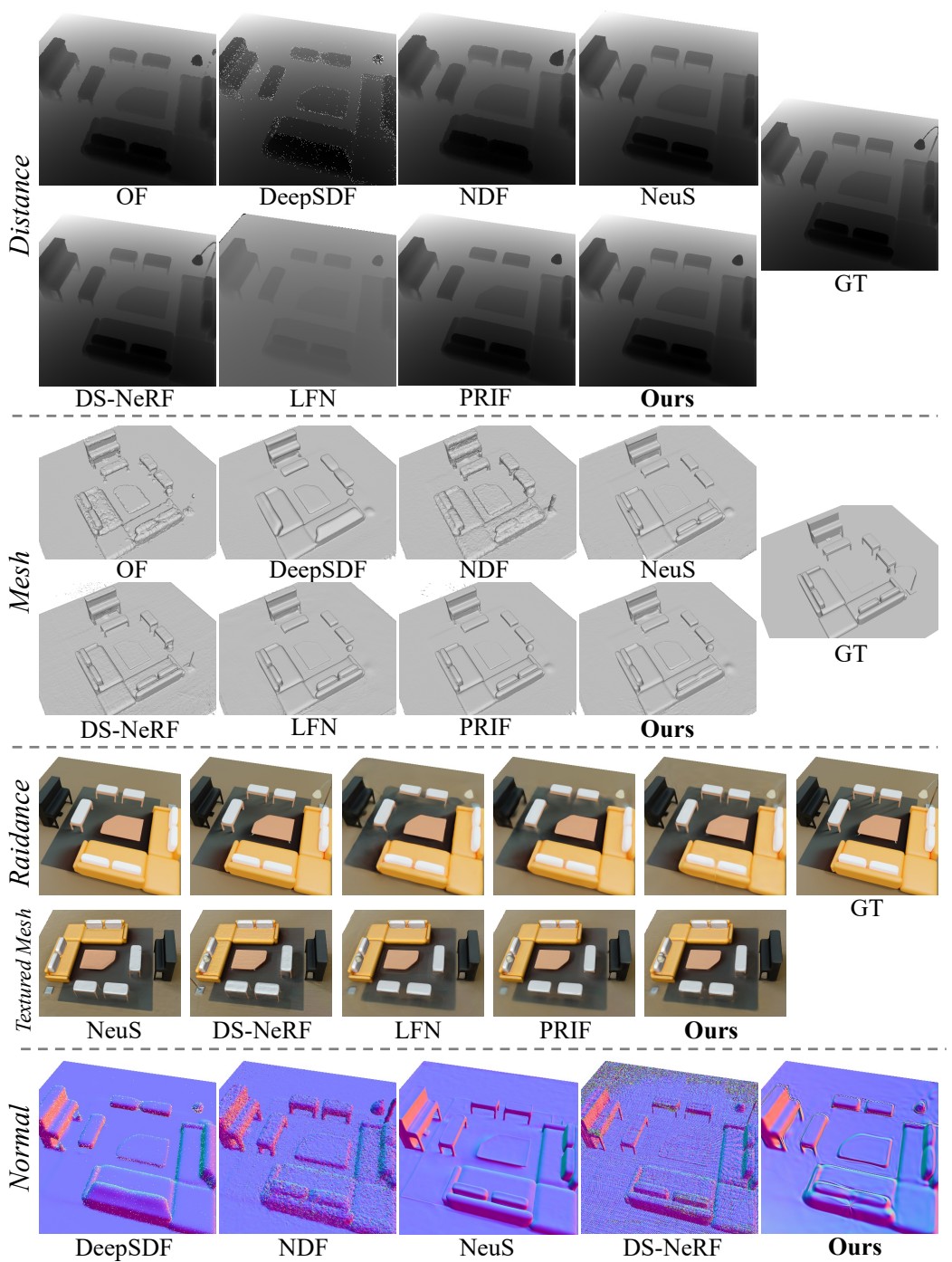

Figure 18: Qualitative results of all methods on *Reception* in DM-SR dataset.

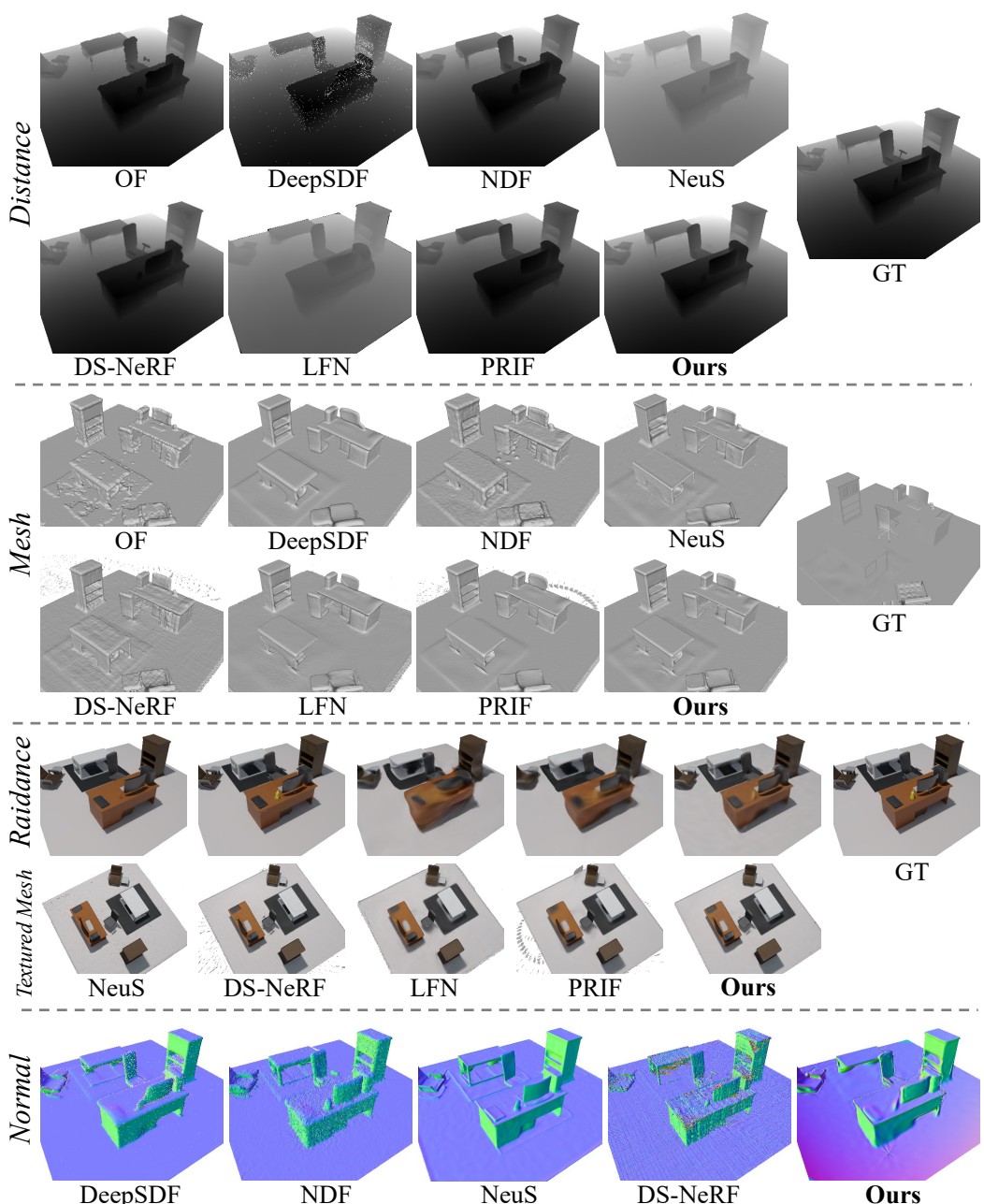

Figure 19: Qualitative results of all methods on *Study* in DM-SR dataset.

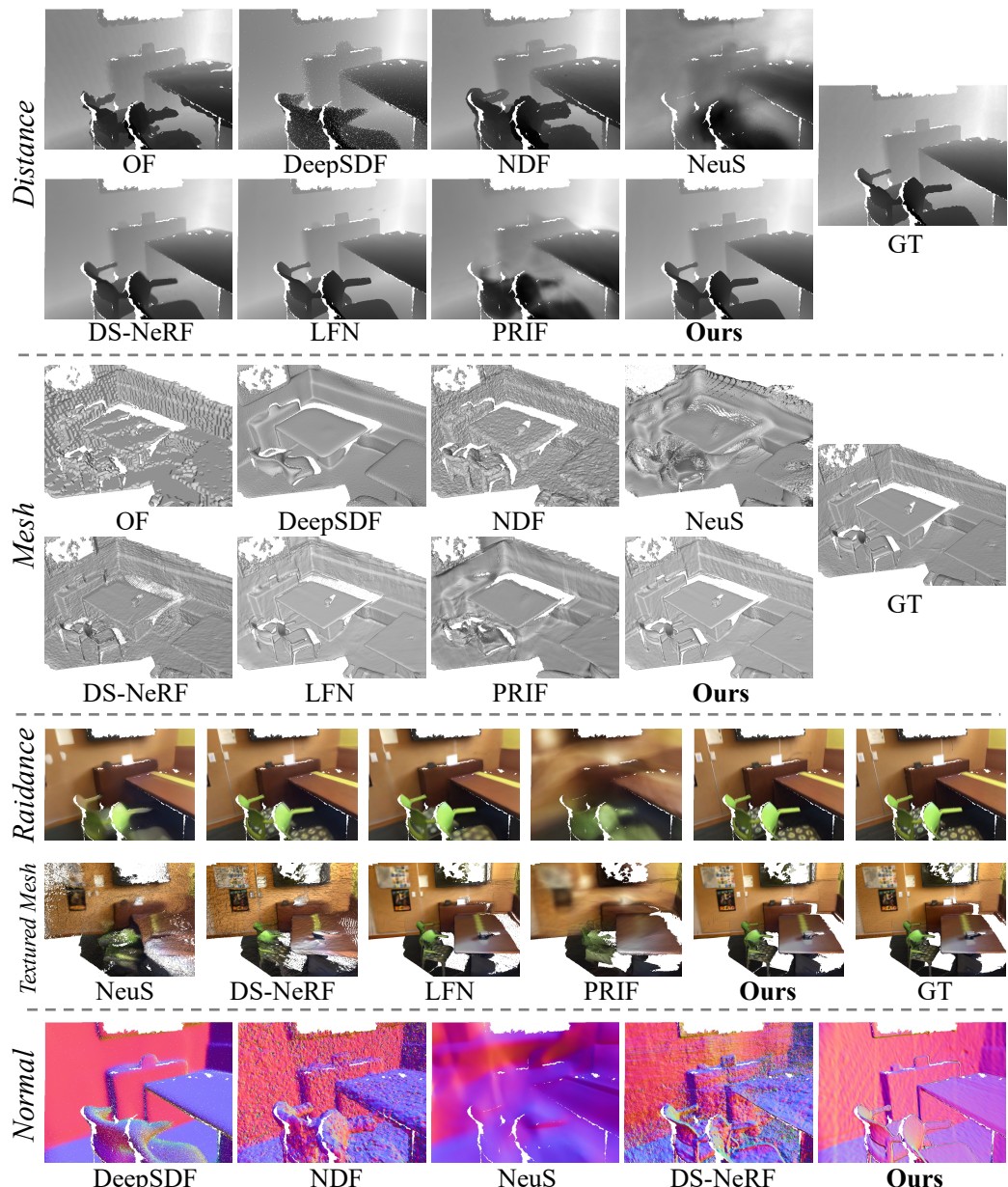

Figure 20: Qualitative results of all methods on *Scene0005_00* in ScanNet dataset.

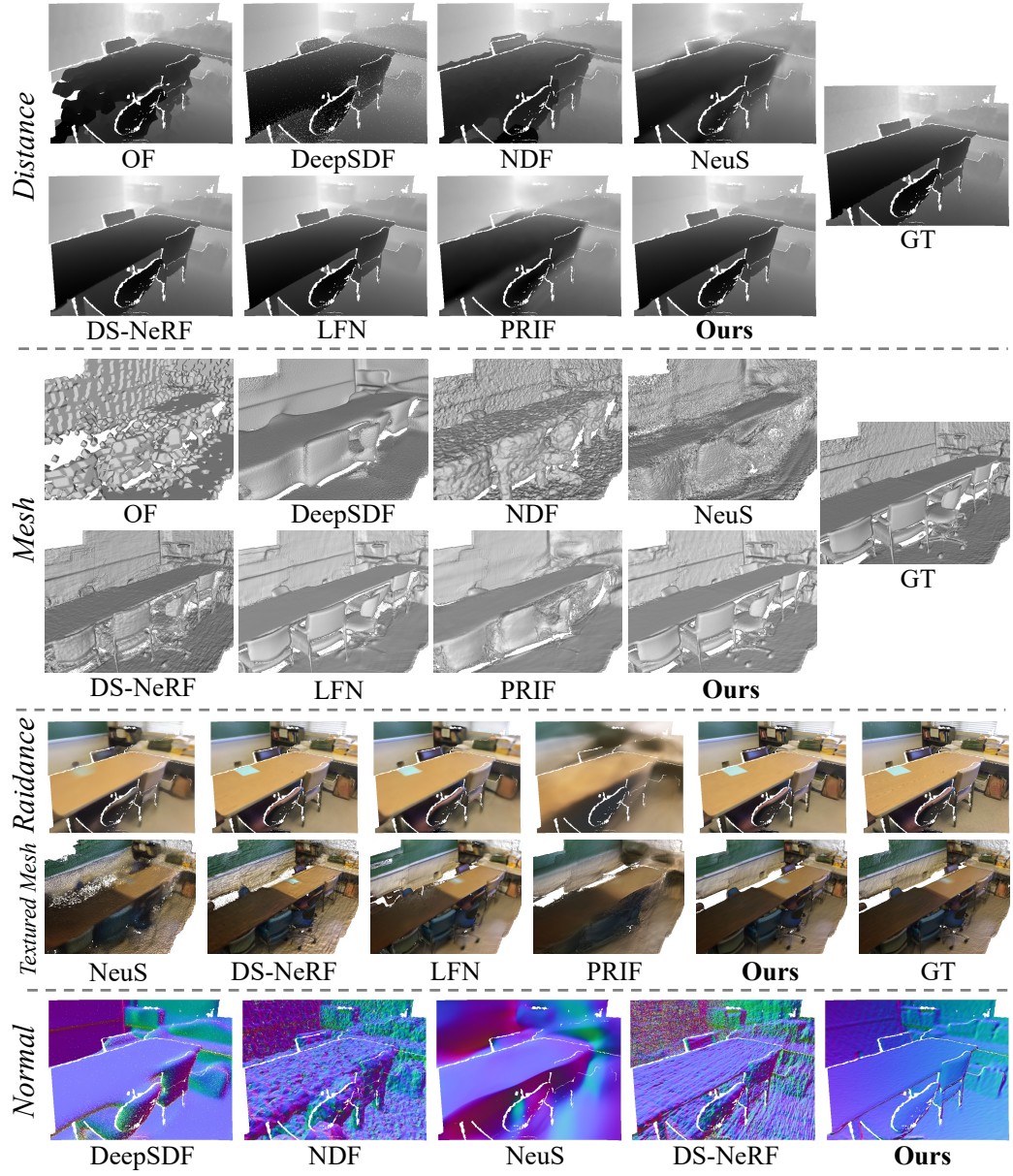

Figure 21: Qualitative results of all methods on *Scene0030_00* in ScanNet dataset.

Table 28: ADE↓ scores of all baselines and our method in experiments of Group 1 on the 8 scenes of Blender dataset.

| | Chair | Drums | Ficus | Hotdog | Lego | Materials | Mic | Ship | Avg. |
|---|---|---|---|---|---|---|---|---|---|
| OF [46] | 6.52 | 14.37 | 19.88 | 4.35 | 9.85 | 6.06 | 8.36 | 15.20 | 10.57 |
| DeepSDF [54] | 6.84 | 16.55 | 22.80 | 8.37 | 12.18 | 9.45 | 9.31 | 18.07 | 12.95 |
| NDF [14] | 7.83 | 14.93 | 17.26 | 7.92 | 11.42 | 11.61 | 9.77 | 16.39 | 12.14 |
| NeuS [77] | 6.87 | 10.61 | 19.02 | 3.36 | 8.43 | 7.43 | 13.92 | 25.37 | 11.88 |
| DS-NeRF [19] | 11.73 | 19.42 | 18.75 | **2.22** | 11.01 | 5.77 | 19.20 | 17.66 | 13.22 |
| LFN [64] | 16.33 | 28.09 | 31.56 | 14.02 | 20.27 | 36.93 | 20.89 | 27.65 | 24.47 |
| PRIF [23] | 7.65 | 19.57 | 25.73 | 5.83 | 12.86 | 15.69 | 13.35 | 16.72 | 14.68 |
| **RayDF (Ours)** | **4.59** | **9.24** | **12.56** | 3.06 | **7.98** | **5.68** | **7.55** | **13.07** | **7.97** |

Table 29: CD ($\times 10^{-3}$)↓ (mean / median) scores of all baselines and our method in experiments of Group 1 on the 8 scenes of Blender dataset.

| | Chair | Drums | Ficus | Hotdog | Lego | Materials | Mic | Ship | Avg. |
|---|---|---|---|---|---|---|---|---|---|
| OF [46] | 4.355 / 0.609 | 2.488 / 1.225 | 2.361 / 0.510 | 8.043 / 0.733 | 1.491 / 0.742 | 0.790 / 0.435 | 1.508 / 0.544 | 2.823 / 0.853 | 2.982 / 0.706 |
| DeepSDF [54] | 4.253 / 0.555 | 2.324 / 1.181 | 5.680 / 0.554 | 8.042 / 0.729 | 1.460 / 0.736 | 0.752 / **0.389** | **1.435** / 0.495 | 3.109 / **0.792** | 3.382 / 0.679 |
| NDF [14] | **1.754** / 0.934 | **2.137** / 1.102 | **2.215** / 0.673 | 10.433 / 0.871 | 1.754 / 0.934 | 1.082 / 0.541 | 1.615 / 0.605 | **2.821** / 0.991 | **2.976** / 0.831 |
| NeuS [77] | 3.831 / 0.372 | 2.582 / 0.885 | 3.376 / 0.920 | 8.022 / **0.630** | 1.282 / 0.665 | 1.771 / 0.918 | 12.238 / 1.816 | 4.948 / 1.053 | 4.756 / 0.907 |
| DS-NeRF [19] | 220.846 / 0.683 | 207.933 / 2.032 | 4.471 / 0.900 | 8.822 / 0.713 | 258.896 / 0.955 | **0.735** / 0.441 | 205.93 / 1.556 | 30.529 / 1.801 | 117.270 / 1.135 |
| LFN [64] | 64.599 / 0.517 | 84.757 / 25.252 | 182.662 / 11.008 | 21.466 / 2.894 | 65.227 / 3.671 | 132.122 / 68.971 | 84.327 / 8.184 | 80.24 / 4.952 | 89.425 / 15.681 |
| PRIF [23] | 12.528 / 0.416 | 21.911 / 4.827 | 20.889 / **0.502** | 22.712 / 1.091 | 12.088 / 0.856 | 27.121 / 2.129 | 24.370 / 2.664 | 24.489 / 0.930 | 20.764 / 1.677 |
| **RayDF (Ours)** | 3.807 / **0.312** | 3.310 / **0.958** | 2.914 / 0.543 | **7.255** / 0.733 | **1.095** / 0.702 | 1.604 / 0.617 | 3.571 / 0.644 | 3.544 / **0.792** | 3.388 / **0.663** |

Table 30: ADE↓ scores of all baselines and our method in experiments of Group 2 on the 8 scenes of Blender dataset.

| | Chair | Drums | Ficus | Hotdog | Lego | Materials | Mic | Ship | Avg. |
|---|---|---|---|---|---|---|---|---|---|
| NeuS [77] | 7.15 | 10.97 | 18.78 | 3.28 | 7.97 | 7.53 | 13.95 | 27.13 | 12.10 |
| DS-NeRF [19] | 9.81 | 15.47 | 18.02 | 6.72 | 10.90 | 18.77 | 19.78 | 17.62 | 14.64 |
| LFN [64] | 6.70 | 15.46 | 20.28 | 4.40 | 13.67 | 12.35 | 10.85 | 14.95 | 12.33 |
| PRIF [23] | 7.51 | 19.20 | 25.25 | 5.37 | 13.02 | 16.24 | 13.29 | 16.57 | 14.56 |
| **RayDF (Ours)** | **4.92** | **9.46** | **12.85** | **2.96** | **8.11** | **6.23** | **7.54** | **13.25** | **8.17** |

Table 31: CD ($\times 10^{-3}$)↓ (mean / median) scores of all baselines and our method in experiments of Group 2 on the 8 scenes of Blender dataset.

| | Chair | Drums | Ficus | Hotdog | Lego | Materials | Mic | Ship | Avg. |
|---|---|---|---|---|---|---|---|---|---|
| NeuS [77] | 3.817 / 0.363 | **2.655** / 0.836 | 3.220 / 0.849 | 7.562 / **0.641** | **1.175** / **0.666** | 1.658 / 0.909 | 11.786 / 1.913 | 5.421 / 1.056 | 4.662 / 0.938 |
| DS-NeRF [19] | 295.445 / 1.931 | 142.057 / 2.192 | **2.969** / 0.775 | 214.536 / 1.603 | 209.45 / 0.976 | 29.646 / 2.072 | 4.697 / 0.725 | 247.558 / 3.802 | 143.295 / 1.760 |
| LFN [64] | 65.602 / 0.516 | 8.036 / 1.962 | 37.458 / **0.480** | 14.783 / 0.753 | 10.065 / 0.992 | 71.93 / 1.367 | 134.494 / 2.648 | 29.944 / 1.122 | 60.289 / 1.230 |
| PRIF [23] | 14.612 / 0.496 | 20.627 / 4.971 | 20.991 / 0.507 | 20.297 / 0.889 | 12.963 / 0.897 | 26.497 / 2.308 | 24.979 / 2.441 | 29.265 / 1.037 | 21.279 / 1.693 |
| **RayDF (Ours)** | **3.629** / **0.306** | 3.433 / 1.278 | 3.419 / 0.587 | **7.253** / 0.733 | 1.372 / 0.802 | **1.551** / 0.659 | **1.798** / 0.813 | **3.907** / 0.858 | **3.295** / 0.755 |

Table 32: PSNR↑ scores of all baselines and our method in experiments of Group 2 on the 8 scenes of Blender dataset.

| | Chair | Drums | Ficus | Hotdog | Lego | Materials | Mic | Ship | Avg. |
|---|---|---|---|---|---|---|---|---|---|
| NeuS [77] | 27.47 | 23.23 | **30.11** | **32.82** | 25.85 | **24.97** | **29.81** | 23.24 | **27.19** |
| DS-NeRF [19] | **29.77** | **23.38** | 23.02 | 32.80 | **28.58** | 23.21 | 28.11 | 24.18 | 26.63 |
| LFN [64] | 25.30 | 19.96 | 21.04 | 29.69 | 21.58 | 20.79 | 26.32 | 20.95 | 23.20 |
| PRIF [23] | 24.72 | 20.07 | 22.69 | 28.85 | 21.65 | 20.00 | 26.01 | 22.46 | 23.31 |
| **RayDF (Ours)** | 27.56 | 23.30 | 29.16 | 31.58 | 25.92 | 22.03 | 28.32 | **24.30** | 26.52 |

Table 33: SSIM↑ scores of all baselines and our method in experiments of Group 2 on the 8 scenes of Blender dataset.

| | Chair | Drums | Ficus | Hotdog | Lego | Materials | Mic | Ship | Avg. |
|---|---|---|---|---|---|---|---|---|---|
| NeuS [77] | 0.914 | 0.911 | 0.968 | 0.957 | 0.873 | 0.900 | **0.966** | 0.800 | 0.910 |
| DS-NeRF [19] | **0.947** | **0.931** | **0.976** | **0.963** | **0.934** | **0.940** | **0.966** | 0.805 | **0.933** |
| LFN [64] | 0.890 | 0.890 | 0.963 | 0.948 | 0.815 | 0.855 | 0.954 | 0.785 | 0.888 |
| PRIF [23] | 0.878 | 0.866 | 0.934 | 0.933 | 0.807 | 0.847 | 0.950 | 0.780 | 0.874 |
| **RayDF (Ours)** | 0.925 | 0.915 | 0.966 | 0.959 | 0.877 | 0.872 | 0.961 | **0.810** | 0.910 |

Table 34: LPIPS↓ scores of all baselines and our method in experiments of Group 2 on the 8 scenes of Blender dataset.

| | Chair | Drums | Ficus | Hotdog | Lego | Materials | Mic | Ship | Avg. |
|---|---|---|---|---|---|---|---|---|---|
| NeuS [77] | 0.108 | 0.096 | 0.029 | 0.068 | 0.114 | 0.080 | **0.037** | 0.250 | 0.100 |
| DS-NeRF [19] | **0.055** | **0.058** | **0.016** | **0.044** | **0.036** | **0.034** | **0.037** | **0.223** | **0.063** |
| LFN [64] | 0.140 | 0.117 | 0.032 | 0.074 | 0.187 | 0.117 | 0.063 | 0.267 | 0.125 |
| PRIF [23] | 0.157 | 0.181 | 0.075 | 0.090 | 0.207 | 0.149 | 0.062 | 0.292 | 0.152 |
| **RayDF (Ours)** | 0.100 | 0.093 | 0.030 | 0.063 | 0.125 | 0.103 | 0.039 | 0.239 | 0.099 |

Table 35: ADE↓ scores of all baselines and our method in experiments of Group 1 on the 8 scenes of DM-SR dataset.

| | Bathroom | Bedroom | Dinning | Kitchen | Office | Reception | Restroom | Study | Avg. |
|---|---|---|---|---|---|---|---|---|---|
| OF [46] | 14.15 | 10.22 | 14.44 | 16.09 | 16.44 | 12.50 | 27.48 | 15.28 | 15.83 |
| DeepSDF [54] | 15.61 | 11.25 | 15.09 | 16.98 | 17.01 | 15.92 | 26.88 | 17.01 | 16.97 |
| NDF [14] | 19.27 | 12.02 | 22.24 | 23.92 | 21.91 | 20.53 | 37.93 | 21.44 | 22.41 |
| NeuS [77] | 7.77 | 4.34 | 9.57 | 12.12 | 6.94 | 17.43 | 13.08 | 8.28 | 9.94 |
| DS-NeRF [19] | 11.25 | **3.04** | 8.83 | 13.13 | 13.41 | 7.51 | 18.93 | 10.03 | 10.77 |
| LFN [64] | 14.71 | 16.16 | 8.11 | 25.07 | 16.58 | 17.20 | 23.88 | 24.65 | 18.30 |
| PRIF [23] | 12.06 | 5.06 | 10.22 | 14.10 | 12.74 | 10.15 | 18.71 | 12.10 | 11.89 |
| **RayDF (Ours)** | **6.24** | 3.79 | **5.30** | **10.63** | **7.08** | **6.96** | **11.44** | **7.81** | **7.41** |

Table 36: CD ($\times 10^{-3}$)↓ (mean / median) scores of all baselines and our method in experiments of Group 1 on the 8 scenes of DM-SR dataset.

| | Bathroom | Bedroom | Dinning | Kitchen | Office | Reception | Restroom | Study | Avg. |
|---|---|---|---|---|---|---|---|---|---|
| OF [46] | 7.481 / 3.285 | 14.923 / 4.509 | **4.305** / 2.581 | 16.764 / **6.271** | 8.939 / 3.405 | **9.391** / 5.111 | 19.926 / 8.875 | **9.484** / 5.066 | 11.402 / 4.888 |
| DeepSDF [54] | 6.929 / 3.441 | **14.601** / 4.888 | 4.555 / 2.506 | **15.881** / 7.119 | 8.768 / 3.763 | 9.998 / 5.070 | **19.524** / 8.405 | 9.989 / 5.504 | **11.281** / 5.087 |
| NDF [14] | **7.096** / 4.299 | 15.821 / 4.784 | 4.386 / 3.102 | 17.745 / 7.737 | 9.745 / 4.472 | 10.468 / 5.432 | 22.141 / 10.802 | 10.999 / 6.662 | 12.300 / 5.911 |
| NeuS [77] | 7.524 / **2.814** | 17.435 / **3.991** | 4.968 / 2.573 | 18.510 / 6.408 | **8.361** / 3.205 | 11.481 / **4.945** | 21.992 / **8.144** | 11.684 / **4.880** | 12.744 / **4.620** |
| DS-NeRF [19] | 9.658 / 3.691 | 16.893 / 4.418 | 5.083 / 2.624 | 36.237 / 12.072 | 27.548 / 4.034 | 28.864 / 5.510 | 46.375 / 9.330 | 32.382 / 6.576 | 25.380 / 6.032 |
| LFN [64] | 8.640 / 3.514 | 18.880 / 4.830 | 5.037 / 2.687 | 20.162 / 7.927 | 9.717 / 3.710 | 10.929 / 5.739 | 21.077 / 8.654 | 14.939 / 5.914 | 13.673 / 5.372 |
| PRIF [23] | 16.864 / 3.382 | 25.057 / 4.437 | 19.999 / **2.437** | 24.584 / 7.444 | 33.328 / 3.749 | 26.009 / 5.160 | 32.373 / 8.452 | 29.732 / 6.212 | 25.993 / 5.159 |
| **RayDF (Ours)** | 7.499 / 3.140 | 17.766 / 4.755 | 4.926 / 2.773 | 18.463 / 7.639 | 9.414 / 3.628 | 10.632 / 5.714 | 20.698 / 8.785 | 24.777 / 6.389 | 14.272 / 5.353 |

Table 37: ADE↓ scores of all baselines and our method in experiments of Group 2 on the 8 scenes of DM-SR dataset.

| | Bathroom | Bedroom | Dinning | Kitchen | Office | Reception | Restroom | Study | Avg. |
|---|---|---|---|---|---|---|---|---|---|
| NeuS [77] | 11.31 | 4.87 | 9.75 | 14.23 | 8.94 | 17.32 | 17.43 | 9.45 | 11.66 |
| DS-NeRF [19] | 11.14 | **3.48** | 8.36 | 13.83 | 13.10 | **7.75** | 18.92 | 9.56 | 10.77 |
| LFN [64] | 13.90 | 16.21 | 7.98 | 24.92 | 17.36 | 19.25 | 24.02 | 24.47 | 18.51 |
| PRIF [23] | 12.02 | 5.06 | 9.32 | 13.96 | 12.54 | 10.21 | 18.85 | 12.18 | 11.77 |
| **RayDF (Ours)** | **6.97** | 4.29 | **5.65** | **11.25** | **7.14** | 8.30 | **11.56** | **8.60** | **7.97** |

Table 38: CD ($\times 10^{-3}$)↓ (mean / median) scores of all baselines and our method in experiments of Group 2 on the 8 scenes of DM-SR dataset.

| | Bathroom | Bedroom | Dinning | Kitchen | Office | Reception | Restroom | Study | Avg. |
|---|---|---|---|---|---|---|---|---|---|
| NeuS [77] | **7.389** / **2.938** | 18.841 / **4.064** | 4.844 / 2.401 | 33.349 / 11.617 | **8.665** / 3.238 | 11.375 / **4.921** | 23.56 / 8.307 | **12.11** / **4.978** | 15.017 / 5.308 |
| DS-NeRF [19] | 9.772 / 3.534 | **17.096** / 4.385 | **4.800** / 2.476 | 40.094 / 13.066 | 38.091 / 4.442 | 26.400 / 5.244 | 35.482 / 9.456 | 32.650 / 6.210 | 25.548 / 6.102 |
| LFN [64] | 8.534 / 3.325 | 19.550 / 4.791 | 4.984 / 2.762 | 21.543 / 8.158 | 10.024 / 3.673 | 11.081 / 5.720 | **20.770** / 8.516 | 16.191 / 5.844 | **14.085** / 5.349 |
| PRIF [23] | 23.415 / 3.328 | 23.450 / 4.302 | 24.105 / **2.368** | 24.322 / 8.452 | 27.253 / 3.434 | 24.142 / 5.386 | 29.527 / **8.167** | 22.524 / 5.813 | 24.842 / **5.156** |
| **RayDF (Ours)** | 7.596 / 3.137 | 17.220 / 4.406 | 4.851 / 2.698 | **18.586** / 7.737 | 9.213 / 3.642 | **10.589** / 5.673 | 21.189 / 8.879 | 24.766 / 6.227 | 14.251 / 5.300 |

Table 39: PSNR↑ scores of all baselines and our method in experiments of Group 2 on the 8 scenes of DM-SR dataset.

| | Bathroom | Bedroom | Dinning | Kitchen | Office | Reception | Restroom | Study | Avg. |
|---|---|---|---|---|---|---|---|---|---|
| NeuS [77] | **33.73** | **37.71** | 29.98 | **33.97** | **36.21** | 30.69 | 31.08 | **32.39** | **33.22** |
| DS-NeRF [19] | 31.26 | 36.67 | **32.36** | 27.73 | 32.78 | **32.26** | 31.02 | 30.56 | 31.83 |
| LFN [64] | 30.77 | 33.82 | 32.29 | 30.48 | 32.61 | 29.47 | 29.08 | 28.35 | 30.86 |
| PRIF [23] | 31.15 | 35.09 | 31.62 | 30.89 | 32.71 | 29.89 | 28.19 | 28.57 | 31.01 |
| **RayDF (Ours)** | 27.98 | 31.07 | 31.23 | 25.03 | 35.91 | 29.27 | **31.47** | 30.57 | 30.32 |

Table 40: SSIM↑ scores of all baselines and our method in experiments of Group 2 on the 8 scenes of DM-SR dataset.

| | Bathroom | Bedroom | Dinning | Kitchen | Office | Reception | Restroom | Study | Avg. |
|---|---|---|---|---|---|---|---|---|---|
| NeuS [77] | 0.972 | 0.978 | 0.919 | **0.973** | 0.981 | 0.966 | 0.965 | 0.963 | 0.965 |
| DS-NeRF [19] | **0.982** | **0.989** | **0.967** | 0.958 | **0.982** | **0.981** | **0.979** | **0.976** | **0.977** |
| LFN [64] | 0.946 | 0.953 | 0.879 | 0.941 | 0.954 | 0.926 | 0.924 | 0.913 | 0.930 |
| PRIF [23] | 0.951 | 0.961 | 0.875 | 0.949 | 0.955 | 0.930 | 0.917 | 0.925 | 0.933 |
| **RayDF (Ours)** | 0.946 | 0.967 | 0.886 | 0.940 | 0.963 | 0.934 | 0.936 | 0.945 | 0.940 |

Table 41: LPIPS↓ scores of all baselines and our method in experiments of Group 2 on the 8 scenes of DM-SR dataset.

| | Bathroom | Bedroom | Dinning | Kitchen | Office | Reception | Restroom | Study | Avg. |
|---|---|---|---|---|---|---|---|---|---|
| NeuS [77] | 0.038 | 0.037 | 0.126 | **0.052** | 0.028 | 0.052 | 0.054 | 0.043 | 0.054 |
| DS-NeRF [19] | **0.018** | **0.011** | **0.030** | 0.073 | **0.026** | **0.016** | **0.018** | **0.018** | **0.026** |
| LFN [64] | 0.088 | 0.095 | 0.201 | 0.099 | 0.064 | 0.107 | 0.110 | 0.122 | 0.111 |
| PRIF [23] | 0.083 | 0.080 | 0.212 | 0.092 | 0.069 | 0.112 | 0.131 | 0.111 | 0.111 |
| **RayDF (Ours)** | 0.092 | 0.078 | 0.217 | 0.107 | 0.083 | 0.113 | 0.121 | 0.093 | 0.113 |

Table 42: ADE↓ scores of all baselines and our method in experiments of Group 1 on 6 scenes of ScanNet dataset.

| | Scene0004_00 | Scene0005_00 | Scene0009_00 | Scene0010_00 | Scene0030_00 | Scene0031_00 | Avg. |
|---|---|---|---|---|---|---|---|
| OF [46] | 26.29 | 13.61 | 13.15 | 11.76 | 23.26 | 6.97 | 15.84 |
| DeepSDF [54] | 19.47 | 11.86 | 10.64 | 9.92 | 14.60 | 5.94 | 12.07 |
| NDF [14] | 31.13 | 20.47 | 21.89 | 17.23 | 29.38 | 12.79 | 22.15 |
| NeuS [77] | 25.75 | 15.76 | 17.79 | 15.58 | 17.50 | 10.82 | 17.20 |
| DS-NeRF [19] | 10.85 | 6.01 | 6.42 | 7.11 | 5.67 | **3.75** | 6.64 |
| LFN [64] | 10.50 | 5.22 | 6.07 | 6.90 | 5.58 | 4.90 | 6.53 |
| PRIF [23] | 13.98 | 9.36 | 9.66 | 9.17 | 12.54 | 7.21 | 10.32 |
| **RayDF (Ours)** | **8.66** | **4.26** | **5.33** | **5.29** | **4.50** | 4.46 | **5.42** |

Table 43: CD ($\times 10^{-3}$)↓ (mean / median) scores of all baselines and our method in experiments of Group 1 on 6 scenes of ScanNet dataset.

| | Scene0004_00 | Scene0005_00 | Scene0009_00 | Scene0010_00 | Scene0030_00 | Scene0031_00 | Avg. |
|---|---|---|---|---|---|---|---|
| OF [46] | **15.039** / 5.642 | 11.443 / 1.655 | 5.157 / 2.638 | 1.436 / 0.841 | 13.921 / 2.364 | 6.949 / 3.721 | 8.991 / 2.810 |
| DeepSDF [54] | 65.574 / 14.168 | 11.173 / 1.754 | 7.344 / 2.618 | 1.691 / 0.824 | 9.869 / 1.986 | 7.548 / 2.842 | 17.200 / 6.255 |
| NDF [14] | 88.485 / 27.523 | 5.893 / 3.227 | 13.861 / 6.474 | 2.973 / 1.400 | 20.873 / 5.331 | 13.503 / 6.130 | 30.436 / 8.348 |
| NeuS [77] | 47.561 / 9.331 | 55.437 / 2.791 | 25.592 / 3.288 | 15.536 / 1.809 | 26.199 / 2.500 | 12.526 / 2.908 | 30.475 / 3.771 |
| DS-NeRF [19] | 28.287 / 6.293 | 8.407 / 1.283 | 6.720 / 2.044 | 2.014 / 0.842 | **9.085** / 1.401 | **3.407** / 2.047 | 9.653 / 2.318 |
| LFN [64] | 17.777 / **4.443** | 5.931 / 1.125 | **3.921 / 1.699** | 3.256 / 0.704 | 10.824 / 1.189 | 4.194 / 2.026 | **7.651** / 1.864 |
| PRIF [23] | 62.527 / 7.605 | 15.450 / 1.926 | 9.499 / 2.386 | 6.030 / 0.909 | 12.951 / 1.557 | 11.515 / 2.583 | 19.662 / 2.828 |
| **RayDF (Ours)** | 33.872 / 4.804 | **5.755 / 0.963** | 4.863 / 1.724 | **0.841 / 0.567** | 11.931 / **1.100** | 5.842 / **1.930** | 10.517 / **1.848** |

Table 44: ADE↓ scores of all baselines and our method in experiments of Group 2 on 6 scenes of ScanNet dataset.

| | Scene0004_00 | Scene0005_00 | Scene0009_00 | Scene0010_00 | Scene0030_00 | Scene0031_00 | Avg. |
|---|---|---|---|---|---|---|---|
| NeuS [77] | 33.43 | 24.05 | 24.18 | 25.84 | 26.57 | 10.67 | 24.12 |
| DS-NeRF [19] | 12.18 | 7.99 | 7.01 | 8.00 | 6.87 | 5.00 | 7.84 |
| LFN [64] | 10.75 | 5.49 | 6.17 | 7.34 | 5.87 | 5.00 | 6.77 |
| PRIF [23] | 13.95 | 9.98 | 9.17 | 9.20 | 14.63 | 6.98 | 10.65 |
| **RayDF (Ours)** | **8.49** | **4.16** | **5.22** | **5.14** | **4.48** | **4.39** | **5.31** |

Table 45: CD ($\times 10^{-3}$)↓ (mean / median) scores of all baselines and our method in experiments of Group 2 on 6 scenes of ScanNet dataset.

| | Scene0004_00 | Scene0005_00 | Scene0009_00 | Scene0010_00 | Scene0030_00 | Scene0031_00 | Avg. |
|---|---|---|---|---|---|---|---|
| NeuS [77] | 73.825 / 11.200 | 109.468 / 8.325 | 46.224 / 4.593 | 27.426 / 2.857 | 44.060 / 3.475 | 104.02 / 3.500 | 67.504 / 5.658 |
| DS-NeRF [19] | 39.896 / 7.466 | 6.186 / 1.487 | 7.052 / 2.171 | 2.170 / 0.992 | 11.084 / 1.484 | 5.434 / 2.638 | 14.642 / 2.950 |
| LFN [64] | **18.970 / 4.586** | **3.838** / 1.104 | **3.901 / 1.697** | 2.882 / 0.700 | 157.685 / 2.838 | **4.290** / 2.048 | 31.928 / 2.162 |
| PRIF [23] | 69.959 / 7.888 | 22.100 / 1.412 | 10.498 / 2.272 | 6.226 / 0.948 | 16.469 / 1.679 | 12.162 / 2.622 | 22.902 / 2.804 |
| **RayDF (Ours)** | 35.163 / 5.502 | 6.273 / **0.988** | 5.658 / 1.782 | **0.945 / 0.590** | **10.231 / 1.150** | 6.165 / **2.040** | **10.739 / 2.009** |

Table 46: PSNR↑ scores of all baselines and our method in experiments of Group 2 on 6 scenes of ScanNet dataset.

| | Scene0004_00 | Scene0005_00 | Scene0009_00 | Scene0010_00 | Scene0030_00 | Scene0031_00 | Avg. |
|---|---|---|---|---|---|---|---|
| NeuS [77] | 28.01 | 24.11 | 27.91 | 25.62 | 26.06 | **30.29** | 27.00 |
| DS-NeRF [19] | 24.50 | 24.42 | 25.26 | 24.70 | 25.18 | 26.11 | 25.03 |
| LFN [64] | 26.91 | 28.71 | 29.12 | 27.96 | 28.27 | 27.87 | 28.14 |
| PRIF [23] | 20.83 | 20.64 | 22.45 | 20.03 | 21.06 | 21.80 | 21.14 |
| **RayDF (Ours)** | **30.61** | **33.48** | **31.31** | **32.10** | **32.37** | 29.59 | **31.58** |

Table 47: SSIM↑ scores of all baselines and our method in experiments of Group 2 on 6 scenes of ScanNet dataset.

| | Scene0004_00 | Scene0005_00 | Scene0009_00 | Scene0010_00 | Scene0030_00 | Scene0031_00 | Avg. |
|---|---|---|---|---|---|---|---|
| NeuS [77] | 0.826 | 0.837 | 0.769 | 0.833 | 0.806 | 0.831 | 0.817 |
| DS-NeRF [19] | **0.861** | 0.889 | 0.796 | 0.884 | 0.862 | **0.839** | 0.855 |
| LFN [64] | 0.794 | 0.872 | 0.770 | 0.858 | 0.817 | 0.775 | 0.814 |
| PRIF [23] | 0.724 | 0.785 | 0.711 | 0.729 | 0.731 | 0.700 | 0.731 |
| **RayDF (Ours)** | 0.848 | **0.912** | **0.797** | **0.903** | **0.870** | 0.805 | **0.856** |

Table 48: LPIPS↓ scores of all baselines and our method in experiments of Group 2 on 6 scenes of ScanNet dataset.

| | Scene0004_00 | Scene0005_00 | Scene0009_00 | Scene0010_00 | Scene0030_00 | Scene0031_00 | Avg. |
|---|---|---|---|---|---|---|---|
| NeuS [77] | 0.266 | 0.225 | 0.347 | 0.197 | 0.209 | 0.270 | 0.253 |
| DS-NeRF [19] | **0.208** | 0.147 | **0.299** | **0.134** | 0.149 | 0.245 | **0.197** |
| LFN [64] | 0.326 | 0.191 | 0.345 | 0.171 | 0.206 | 0.334 | 0.262 |
| PRIF [23] | 0.427 | 0.350 | 0.420 | 0.347 | 0.319 | 0.465 | 0.388 |
| **RayDF (Ours)** | 0.261 | **0.146** | 0.328 | 0.135 | 0.157 | 0.319 | 0.224 |

