# OpenReview forum: "RayDF: Neural Ray-surface Distance Fields with Multi-view Consistency"
_NeurIPS.cc/2023/Conference — NeurIPS 2023 poster_

### Official Review · Reviewer_MPar · 2023-07-05

**Soundness:** 4 excellent
**Presentation:** 4 excellent
**Contribution:** 4 excellent
**Rating:** 6
**Confidence:** 4

**Summary:**

This paper proposes MucRays, which imposes ray-based neural functions with multi-view geometry consistency. This framework contains three-parts: ray-surface distance field, dual-ray visibility classifier, and multi-view consistency optimization strategy. Quantitative results of MucRays surpass the existing coordinate- or ray- based networks for synthetic or real-world scene modeling. This ray-based method achieves 1000x faster for depth image inference.

**Strengths:**

1.	The dual-ray visibility classifier is interesting. It is well-motivated and acts as the foundation of the multi-view consistency optimization.
2.	The quantitative results are good. The high performance of MucRays on DAE metric proves its ability to render more accurate depth images.


**Weaknesses:**

1.	The qualitative results does not satisfy me. It seems that, MucRays has difficulty in representing thin structures. For example, in the Reception scene in Figure 4 (Appendix), the arm of the desk lamp is missing in both distance map and mesh. I would like to see a discussion about such phenomenon.
2.	I think NeuS (NeuS: Learning Neural Implicit Surfaces by Volume Rendering for Multi-view Reconstruction) is a better baseline than DeepSDF because it could use both depth and color for supervision.
3.	I assume that this paper focused on better geometry modeling. However, it seems that the geometric result (mesh result) on Lego (Figure 5) is over smoothed compared with DS-NeRF or NDF. Is it caused by TSDF or the nature of ray distance representation itself?


**Questions:**

Main questions are in Weakness part.

More detailed comments:

The authors should use consistent abbreviation. For example, L. 80 use UDF for paper [13], while the remaining contents use NDF for [13].

**Limitations:**

Yes.

---

> ### Author Rebuttal · Authors · 2023-08-10
>
> We appreciate the reviewer's thoughtful comments and address the main concerns below.
>
> **Q1: The qualitative results does not satisfy me. It seems that, MucRays has difficulty in representing thin structures. For example, in the Reception scene in Figure 4 (Appendix), the arm of the desk lamp is missing in both distance map and mesh. I would like to see a discussion about such phenomenon.**
>
> **A1:** Unarguably, recovering thin structures is particularly challenging for almost all methods, including ours and baselines. For our method, there could be multiple potential factors that lead to difficulty in reconstructing thin structures.
>
> In our multi-view consistency loss function $\ell_{mv}$ (Eq. 3), we simply change it from $\ell_1$ to $\ell_2$ optimization, and conduct an additional experiment on *Reception* of DMSR dataset. The following Table and Figure 8 in the submission file show the results. We can see that $\ell_2$ based loss function can better reconstruct thin structures but with more noisy results at the same time. In this regard, it is of great interest to fully address this issue and we leave it for future exploration.
>
> |                    | DAE$\downarrow$| CD ($\times10^{-3}$) $\downarrow$ (mean / median)  |
> |--------------------|:---------:|:-----------------:|
> | MucRays-$\ell_2$|**6.56** | 12.141 / 6.853   |
> | **MucRays-$\ell_1$** | 6.96| **10.632** / **5.714**|
>
> **Q2: I think NeuS (NeuS: Learning Neural Implicit Surfaces by Volume Rendering for Multi-view Reconstruction) is a better baseline than DeepSDF because it could use both depth and color for supervision.**
>
> **A2:** Thank you for this suggestion and we conduct additional experiments for NeuS on the Blender dataset. From the following Table, our method is clearly better than NeuS in both reconstruction accuracy and novel view rendering speed. Figure 9 in the submission file shows the qualitative results. We will add this new baseline into the main paper in next version.
>
> |                    | DAE$\downarrow$| CD ($\times10^{-3}$) $\downarrow$ (mean / median)  | PSNR $\uparrow$| SSIM $\uparrow$ | LPIPS $\downarrow$ | Rendering time (seconds)
> |--------------------|:---------:|:-----------------:|:---:|:---:|:---:|:---:|
> | NeuS| 12.10 | 4.662 / 0.938   | **27.19** | **0.911** | **0.100** | 32.793 |
> | **MucRays** | **8.17** | **3.295** / **0.755**| 26.52 | 0.910 | 0.099 | **0.019**|
>
> **Q3: I assume that this paper focused on better geometry modeling. However, it seems that the geometric result (mesh result) on Lego (Figure 5) is over smoothed compared with DS-NeRF or NDF. Is it caused by TSDF or the nature of ray distance representation itself?**
>
> **A3:** This is a very interesting question. The observed over-smoothing effects may be caused by multiple potential factors, including the choice of ray parameterizations, the choice of loss functions, the postprocessing of TSDF, etc. Additionally, the nature of continuous ray-distance functions (MLPs) is also likely to result in smoothing predictions given similar input rays. In this regard, we hope that our method could inspire more advanced research works to tackle these core challenges in the future.
>
> **Q4: The authors should use consistent abbreviation. For example, L. 80 use UDF for paper [13], while the remaining contents use NDF for [13].**
>
> **A4:** Thanks. We will fix it in the next version.

---

> > ### Comment · Reviewer_MPar · 2023-08-11
> > **Reply**
> >
> > The rebuttal has solved my concerns and I maintain my initial rating.

---

> > > ### Author Response · Authors · 2023-08-13
> > > **Thanks**
> > >
> > > We thank the reviewer's time to read our rebuttal materials and the encouraging rating. -Authors.

---

### Official Review · Reviewer_sym2 · 2023-07-06

**Soundness:** 3 good
**Presentation:** 4 excellent
**Contribution:** 3 good
**Rating:** 7
**Confidence:** 5

**Summary:**

This paper presents a new strategy to design ray-based neural representation of 3D shapes. Ray-based approaches towards shape representation is a recently emerging idea that bypasses extensive point-based evaluation required by conventional methods like signed distance function. A key missing component in existing ray-based neural representations is the lack of multi-view consistency. This paper proposes training a dual-ray visibility classifier and using that network's output to guide the training of the network for ray-distance prediction. The method is extensively evaluated on synthetic and real-world datasets, showing its superiority over previous ray-based methods in predicting ray hit points from novel views. Moreover, this paper also includes evaluations of color prediction of this ray-based method, showing it achieving comparable performance to previous efficient representations.

**Strengths:**

- Addresses a key issue of multi-view inconsistency in current ray-based scene representations.
- Clear writing and figure presentation.
- Extensive evaluation showing the efficacy of the proposed strategy, outperforming previous baselines without slowing down inference

**Weaknesses:**

- The proposed method operates under the setting where depth maps are available for all views, which is okay, but the paper would be more complete if it discusses how bad the results would become if depth is unknown, which is actually the setting of LFN.
- Not totally clear how the baselines are implemented. For example, when using LFN and PRIF, the supplementary material mentions that it uses the same architecture as the official setup. However, it is not clear whether the LFN/PRIF results are obtained still with the proposed classifier. At least based on the writing, it does not eliminate the ambiguity.
- The proposed method also adopts a different ray parametrization than LFN and PRIF. It is not clear what the reason is behind this design choice.
- The proposed method is implemented as a 13-layer SIREN with 1024 hidden channels, which is different than previous baselines. The paper does not include any analysis on the impact of varying layer depth and hidden channels.
- In L123, the paper says the network predictions "must satisfy a transformation equation", without defining what this equation is.
- Related work misses any discussion on space carving / visual hull (e.g., [Matusik 2000], [Kutulakos 2000]), which are important early concepts behind the idea of visibility checking proposed in this work. Related work also misses some recent papers on light field / ray-based representation (e.g. HyperReel [CVPR 2023], SRT [CVPR 2022], SIGNET [ICCV 2021]).
- L237 typo: "RTX 30390 GPU"

**Questions:**

- In Table 5, the performance drops significantly after removing the classifier, and it actually becomes much worse than the various baselines in Table 2. This is confusing and relates to the above-mentioned weakness about "Not totally clear how the baselines are implemented". Do the baseline methods included in Table 2, specifically LFN and PRIF, involve a classifier? That would totally change how to correctly interpret the results in Table 5.
- If the baseline methods do not involve a classifier, and they are only just about a different ray parametrization with slightly smaller network, then the natural question becomes: how does LFN/PRIF + classifier perform?
- In general, how does the ray parametrization affect the performance? It appears that it would have minor impact compared to the role of classifier, but the paper does not provide enough information to draw any conclusion.

**Limitations:**

Yes

---

> ### Author Rebuttal · Authors · 2023-08-10
>
> We appreciate the reviewer's thoughtful comments and address the main concerns below.
>
> **Q1: The proposed method operates under the setting where depth maps are available for all views, which is okay, but the paper would be more complete if it discusses how bad the results would become if depth is unknown, which is actually the setting of LFN.**
>
> **A1:** We thank the reviewer for this suggestion. Our method indeed operates with the requirement of depth supervision. Nevertheless, with the advancement of existing techniques of depth estimation from RGB images, it is quite feasible to obtain sparse depth signals using existing techniques such as SfM or learning-based monocular depth estimators. In this regard, we additionally provide experimental results of our method using sparse depth values in supervision. From the following Table which reports the metrics on *Lego* in Blender dataset, we can see that our method still achieves satisfactory performance even though there are only 1\% depth values in training. We hypothesize that such robustness comes from our simple multiview consistency constraint, because many depth values in the training set may be redundant thanks to our effective classifier. Figure 4 in the submission file shows the qualitative results. We will include these results in the next version.
>
> |Sparsity of depth supervision| DAE$\downarrow$| CD ($\times10^{-3}$) $\downarrow$ (mean / median)  |
> |--------------------|:---------:|:-----------------:|
> | 1% |  8.69  | 1.517 / 0.937   |
> | 5% |  8.55  | 1.522 / 0.928   |
> | 10% | 8.06  | 1.503 / 0.952   |
> | **100% (MucRays)** | **7.98**| **1.095** / **0.702**|
>
> **Q2: Not totally clear how the baselines are implemented. For example, when using LFN and PRIF, the supplementary material mentions that it uses the same architecture as the official setup. However, it is not clear whether the LFN/PRIF results are obtained still with the proposed classifier. At least based on the writing, it does not eliminate the ambiguity.**
>
> **A2:** The LFN/PRIF results are obtained without our classifier. Here, we further conduct experiments on LFN/PRIF by using our own classifier for comparison. As shown in the following Table, our method still achieves the best performance. Figure 1 in the submission file shows the qualitative results.
>
> |                    | DAE$\downarrow$| CD ($\times10^{-3}$) $\downarrow$ (mean / median)  |
> |--------------------|:---------:|:-----------------:|
> | LFN + Our Visibility Classifier |  90.82  | 121.289 / 44.564   |
> | PRIF + Our Visibility Classifier |  9.97  | 4.187 / 0.891   |
> | **MucRays** | **7.97**| **3.388** / **0.663** |
>
> **Q3: The proposed method also adopts a different ray parametrization than LFN and PRIF. It is not clear what the reason is behind this design choice.**
>
> **A3:** In fact, our pipeline is amenable to different types of ray parameterizations. To verify this, we conduct additional ablation studies by replacing our spherical coordinate with that used in LFN and PRIF. The following Table shows the results on the Blender dataset. We can see that PRIF parameterization can also achieve satisfactory performance, while LFN parameterization is inferior in our pipeline. Figure 1 in the submission file shows qualitative results.
>
> |                    | DAE$\downarrow$| CD ($\times10^{-3}$) $\downarrow$ (mean / median)  |
> |--------------------|:---------:|:-----------------:|
> | LFN param. + (Ray-surface Distance Network + Visibility Classifier) |  89.53  | 82.102 / 10.298   |
> | PRIF param. + (Ray-surface Distance Network + Visibility Classifier) |  8.42  | 3.409 / **0.621**   |
> | (LFN param. + Ray-surface Distance Network) + (Our param. + Visibility Classifier)|  91.30  | 142.570 / 74.345   |
> | (PRIF param. + Ray-surface Distance Network) + (Our param. + Visibility Classifier) |  8.44  | 3.526 / 0.653   |
> | **MucRays** | **7.97**| **3.388** / 0.663|
>
> **Q4: The proposed method is implemented as a 13-layer SIREN with 1024 hidden channels, which is different than previous baselines. The paper does not include any analysis on the impact of varying layer depth and hidden channels.**
>
> **A4:** Thanks for the suggestion to analyze different layers and channels. We conduct additional experiments as shown in the following Table. We can see that our framework tends to favor a wider or deeper network. However, to extensively explore an optimal network architecture is non-trivial and we leave it for our future work.
>
> |                    | DAE$\downarrow$| CD ($\times10^{-3}$) $\downarrow$ (mean / median)  |
> |--------------------|:---------:|:-----------------:|
> | 8 layers, 512 hidden channels |  9.13  | 4.311 / 0.954   |
> | 8 layers, 1024 hidden channels |  8.52  | 4.010 / 0.805   |
> | 13 layers, 512 hidden channels |  8.80  | 4.085 / 0.854   |
> | **MucRays** | **7.97**| **3.388** / **0.663**|
>
> **Q5: In L123, the paper says the network predictions "must satisfy a transformation equation", without defining what this equation is.**
>
> **A5:** Thank you for pointing it out. The transformation equation is originally presented in Appendix A.1.3. and we will move it to the main text in the next version.
>
> **Q6: Related work misses any discussion on space carving / visual hull (e.g., [Matusik 2000], [Kutulakos 2000]), ...... Related work also misses some recent papers on light field / ray-based representation (e.g. HyperReel [CVPR 2023], SRT [CVPR 2022], SIGNET [ICCV 2021]).**
>
> **Q7: L237 typo: "RTX 30390 GPU".**
>
> **A6-A7:** Thank you for sharing the related works. We will include and discuss them in our next version. Typos will be corrected as well.
>
> **Q8: In Table 5, the performance drops significantly after removing the classifier ......**
>
> **A8:** Responded in **Q2**.
>
> **Q9: If the baseline methods do not involve a classifier ......**
>
> **A9:** Responded in **Q2**.
>
> **Q10: In general, how does the ray parametrization affect the performance? ......**
>
> **A10:**  Responded in **Q3**.

---

> ### Comment · Reviewer_sym2 · 2023-08-12
>
> The response from the authors has sufficiently addressed the issues raised in the initial review. The final rating is updated to"Accept".

---

> > ### Author Response · Authors · 2023-08-13
> > **Thanks**
> >
> > We thank the reviewer's time to read our rebuttal materials and the very positive rating. -Authors.

---

### Official Review · Reviewer_4cJL · 2023-07-08

**Soundness:** 3 good
**Presentation:** 3 good
**Contribution:** 2 fair
**Rating:** 6
**Confidence:** 3

**Summary:**

The paper proposed a ray-based neural rendering method that is able to achieve good reconstruction quality from depth maps or RGBD inputs using only one network evaluation per pixel during evaluation.
The method requires two networks: a ray-surface distance network, and a dual-ray visibility classifier. In the first stage, the visibility classifier is trained on the multi-view depth maps. In the second stage, the ray-surface distance network can be trained with both ground truth rays from the depth maps, and random sampled novel rays. the visibilities of the random rays are determined by te visibility classifier, and only the distance of the visible rays are supervised. The proposed approach is extremely fast, and achieves competitive quality on three RGBD datasets.

**Strengths:**

* The proposed two-stage method is novel.
* The proposed method is extremely fast during evaluation -- only one network evaluation is needed per ray.
* The method achieves competitive performance on both synthetic and real datasets.
* The paper provided useful insights on the derivation of surface normals from the proposed ray distance field.

**Weaknesses:**

* The method requires ground truth depth information for training, which is not the case for light field works such as [57]. This can significantly limit its use cases.
* There lacked comparisons with some good performing light field methods such as [57].
* The method relies on the visibility classifier to generalize to novel rays, which can potentially be unreliable.
* Ther performance of the proposed method is lacking, especially according to the results in the appendix.
* It might help understand the effect of visibility network better if there are visualizations of the visibility network predictions.

**Questions:**

There appear to be some fixed-pattern noise artifacts in the demo video. I wonder what is the cause?

**Limitations:**

The limitations and broader impacts of the works are adequately discussed in the paper.

---

> ### Author Rebuttal · Authors · 2023-08-10
>
> We appreciate the reviewer's thoughtful comments and address the main concerns below.
>
> **Q1: The method requires ground truth depth information for training, which is not the case for light field works such as [57]. This can significantly limit its use cases.**
>
> **Q2: There lacked comparisons with some good performing light field methods such as [57].**
>
> **A1-A2:** The primary goal of our method is to model 3D surface geometry, instead of learning radiance fields for novel view RGB rendering. LFNR [57] is an image-conditioned light field method for RGB rendering, which is dramatically different and not directly comparable with ours. Unarguably, both pipelines have their own merits and applications, and our method has great potential in robotics mapping, navigation, obstacle avoidance, etc.
>
> Our method indeed requires depth supervision in the current submission. However, we argue that, with the advancement of existing techniques of depth estimation from RGB images, it is quite feasible to obtain sparse depth signals using existing techniques such as SfM or learning-based monocular depth estimators. In this regard, we additionally provide experimental results of our method using sparse depth values in supervision. From the following Table which reports the metrics on *Lego* in Blender dataset, we can see that our method still achieves satisfactory performance even though there are only 1\% depth values in training. We hypothesize that such robustness comes from our simple multiview consistency constraint, because many depth values in the training set may be redundant thanks to our effective classifier. Figure 4 in the submission file shows the qualitative results. We will include these results in the next version.
>
> |Sparsity of depth supervision| DAE$\downarrow$| CD ($\times10^{-3}$) $\downarrow$ (mean / median)  |
> |--------------------|:---------:|:-----------------:|
> | 1% |  8.69  | 1.517 / 0.937   |
> | 5% |  8.55  | 1.522 / 0.928   |
> | 10% | 8.06  | 1.503 / 0.952   |
> | **100% (MucRays)** | **7.98**| **1.095** / **0.702**|
>
> **Q3: The method relies on the visibility classifier to generalize to novel rays, which can potentially be unreliable.**
>
> **A3:** This is a very interesting point. To further evaluate the reliability of our pipeline, we conduct a series of experiments to add different levels of noise into our classifier. In particular, we directly add a random noise drawn from a normal distribution (0, $\sigma$) to the visibility score with a clip between 0 and 1, and then use the noisy score in our multiview consistency loss $\ell_{mv}$ in Eq. 3 to optimize our ray surface distance network.
>
> The following Table shows the accuracy and F1 scores of our visibility classifier and the final DAE/CD scores on *Lego* in Blender dataset. We can see that, once the accuracy of the visibility classifier degrades, the reconstruction performance decreases sharply due to the large amount of mismatched training signals. This highlights that our classifier plays a crucial role in the pipeline. Figure 5 in the submission file shows qualitative results. We will include these results in the next version.
>
> |Noise level ($\sigma^2$)|Acc. (\%)$\uparrow$ | F1 (\%)$\uparrow$ |DAE$\downarrow$| CD ($\times10^{-3}$) $\downarrow$ (mean / median)  |
> |--------------------|:---------:|:-----------------:|:-----:|:-----:|
> | 1 | 62.56 | 59.06 | 23.09 | 2.713 / 0.960   |
> | 0.5 | 63.99 | 67.76 | 21.16 | 2.248 / 0.889   |
> | 0.1 | 68.01 | 72.59 | 16.83  | 1.850 / 0.860   |
> | **0 (MucRays)** |  **87.74** | **85.24**  | **7.98**| **1.095** / **0.702**|
>
>
> **Q4: The performance of the proposed method is lacking, especially according to the results in the appendix.**
>
> **A4:** Our method is extensively evaluated on three public datasets together with a series of ablation studies to verify the effectiveness of our design, achieving a clear advantage in surface point regression both in terms of accuracy and efficiency. We are always open to conducting more experiments at the request of reviewers.
>
> **Q5: It might help understand the effect of visibility network better if there are visualizations of the visibility network predictions.**
>
> **A5:** We highly appreciate this suggestion. To better understand the classifier, we initially present quantitative results in Table 3 in Appendix. In addition, we further provide qualitative results of the classifier (query point + sampling rays with visibility predictions) in Figure 6 in the submission file, which will be added to the paper in the next version.
>
> **Q6: There appear to be some fixed-pattern noise artifacts in the demo video. I wonder what is the cause?**
>
> **A6:** Thank you for pointing out this issue. The artifacts are caused by the choice of spherical coordinate.
>
> As also suggested by Reviewer sym2, we conduct additional experiments given two different choices of ray parameterizations on the Blender dataset. As shown in the following Table, our applied spherical coordinate can achieve the best DAE scores, and the caused artifacts can be easily fixed by using PRIF's ray parameterization as shown in Figure 7 in the submission file. We leave it for our future work to fully address such an issue.
>
> |                    | DAE$\downarrow$| CD ($\times10^{-3}$) $\downarrow$ (mean / median)  |
> |--------------------|:---------:|:-----------------:|
> | LFN param. + (Ray-surface Distance Network + Visibility Classifier) |  89.53  | 82.102 / 10.298   |
> | PRIF param. + (Ray-surface Distance Network + Visibility Classifier) |  8.42  | 3.409 / **0.621**   |
> | (LFN param. + Ray-surface Distance Network) + (Our param. + Visibility Classifier)|  91.30  | 142.570 / 74.345   |
> | (PRIF param. + Ray-surface Distance Network) + (Our param. + Visibility Classifier) |  8.44  | 3.526 / 0.653   |
> | **MucRays** | **7.97**| **3.388** / 0.663|

---

> > ### Comment · Reviewer_4cJL · 2023-08-14
> > **Thanks for the rebuttal**
> >
> > I would like to thank the authors for the clarifications, as well as the extra experiments and figures. I decide to keep my rating of weak accept. Although the paper has proposed an interesting way to model 3D geometry, its drawbacks are also quite significant: it requires depth maps to train, and the quality does not stand out among baselines.

---

> > > ### Author Response · Authors · 2023-08-15
> > > **Thanks**
> > >
> > > We thank the reviewer's time in reviewing our rebuttal materials and providing valuable feedback. We agree to the desirability of reconstructing geometry solely from RGB images. Nevertheless, considering the range of input modalities (RGB, Depth, Sparse Point Clouds, etc.), along with the requirements for high output surface accuracy and efficient rendering, it is a significant challenge to achieve a good balance.
> > >
> > > While our method indeed utilizes (sparse) depth values in training, it achieves the best reconstruction accuracy over the existing OF/SDF/UDF/NeRF/LFN/Distance based methods under equivalent depth supervision. In addition, it's 1000x faster in rendering views than prevailing coordinate-based methods (OF/SDF/UDF/NeRF).
> > >
> > > There are also challenges remaining, including training without poses, training with RGB alone, generalizing to multi-scenes, enhancing network backbones, expediting training, and more. We hope that our paper could inspire more advanced methods to tackle all these challenges in the future.
> > >
> > > On the whole, we highly appreciate the reviewer's efforts in improving our manuscript and fostering thought-provoking discussions.

---

### Official Review · Reviewer_fezd · 2023-07-10

**Soundness:** 3 good
**Presentation:** 3 good
**Contribution:** 2 fair
**Rating:** 4
**Confidence:** 3

**Summary:**

This paper proposes a framework MucRays for 3D shape representation. Specifically, the authors formulate 3D shapes as ray-based neural functions and incorporate multi-view geometry consistency to improve the performance. For the learning of multi-view geometry consistency, an auxiliary network is introduced to classify the mutual visibility of two sampled rays. Experiments are conducted on various datasets, showing good performance.

**Strengths:**

1. The paper is easy to follow and understand.
2. The authors propose an effective framework for 3D shape representation, showing good performance in various datasets.

**Weaknesses:**

1. This paper only compares rendering time, but training/optimization time is not compared.
2. The ablation study is not convincing. The authors should compare with and without postprocessing (Section 3.2).
3. Table 5, the performance of "w/o classifier" is quite worse than the full model, and the error is very high. It would be better to discuss some possible reasons.
4. 2-stage training is not elegant. Is that possible to train these two networks simultaneously?
5. Line 57, "ray-surface distance field" representation is introduced by previous works and cannot be summarized as one contribution of this work.

**Questions:**

See weaknesses.

**Limitations:**

Negative impacts are mentioned in the paper.

---

> ### Author Rebuttal · Authors · 2023-08-10
>
> We appreciate the reviewer's thoughtful comments and address the main concerns below.
>
> **Q1: This paper only compares rendering time, but training/optimization time is not compared.**
>
> **A1:** Thanks for the suggestion. The following Table compares the average training time of our method and all baselines on each scene of Blender dataset, using a single NVIDIA RTX 3090 GPU card and a CPU of AMD Ryzen 7. Since we apply a two-stage training strategy, our method is not as fast as baselines during optimization. Nevertheless, once our ray-surface distance network is optimized, it can achieve superior efficiency in rendering novel views, as shown in Table 1 in the main text. We will clarify this point in the next version.
>
> |                    | time (hours) |
> |--------------------|:--------------:|
> | OF                 | **0.2**      |
> | DeepSDF            | 0.6          |
> | NDF                | 2.1          |
> | DS-NeRF            | 22.8         |
> | NeuS               | 10.2         |
> | LFN                | 0.9          |
> | PRIF               | 2.2          |
> | **MucRays (Ours)** | 24.9         |
>
> **Q2: The ablation study is not convincing. The authors should compare with and without postprocessing (Section 3.2).**
>
> **A2:** Section 3.2 does not mention any postprocessing step. In Section 3.4, we discuss the removal of potential outlier 3D points with the aid of our derived closed-form surface normals. Note that, this postprocessing step is only applied to clean the reconstructed 3D point clouds when calculating CD scores. We do not use any postprocessing step when calculating all DAE scores. As shown in the following Table, we conduct additional experiments w/ and w/o the outlier removal step on the Blender dataset. Figure 2 in the submission file shows the qualitative results. We will slightly rephrase lines 197-198 in Section 3.4 to clarify how this simple postprocessing step can be used when obtaining explicit 3D point clouds.
>
> |                    | DAE$\downarrow$| CD ($\times10^{-3}$) $\downarrow$ (mean / median)  |
> |--------------------|:---------:|:-----------------:|
> | w/o post-processing|**7.97** | 4.157 / 0.777   |
> | **MucRays (w/ post-processing)** | **7.97**| **3.388** / **0.663**|
>
> **Q3: Table 5, the performance of "w/o classifier" is quite worse than the full model, and the error is very high. It would be better to discuss some possible reasons.**
>
> **A3:** This is a very good point. The key issue of a naive ray based distance function is the lack of generalization ability across novel views during testing, fundamentally because training a single ray based network (without classifier) tends to fit all ray distance data pairs of the training set, but cannot guarantee the consistency of surface distances between seen (training) rays and unseen (testing) rays.
> However, with the help of our well-trained dual ray visibility classifier, the ray distance network must satisfy a transformation equation (Appendix A.1.3). This is easily achieved by our multiview consistency loss function $\ell_{mv}$ (Eq. 3 of main text), which drives the consistency of surface distances between (unlimited) seen and unseen rays.  We show the qualitative comparisons in Figure 3 in the submission file.
>
> **Q4: 2-stage training is not elegant. Is that possible to train these two networks simultaneously?**
>
> **A4:** We agree that an ideal strategy is to train both networks simultaneously. As shown in the following Table and Figure 3 in the submission file, we simply train our two networks at the same time on the Blender dataset. However, not surprisingly, the performance of one-stage training drops noticeably, primarily because the classifier is inaccurate at the early stage and unlikely to provide effective constraints for the ray distance network given the similar number of training steps. Nevertheless, it is an interesting direction for our future work.
>
> |                    | DAE$\downarrow$| CD ($\times10^{-3}$) $\downarrow$ (mean / median)  |
> |--------------------|:---------:|:-----------------:|
> | one-stage training |  12.41  | 4.032 / 0.659   |
> | **two-stage training** | **7.97**| **3.388** / **0.663**|
>
> **Q5: Line 57, "ray-surface distance field" representation is introduced by previous works and cannot be summarized as one contribution of this work.**
>
> **A5:** Thank you for this advice. We will rephrase lines 57-58 or alternatively combine lines 57-60 in the next version.

---

> > ### Author Response · Authors · 2023-08-16
> > **Waiting for Discussion**
> >
> > Dear reviwer fezd,
> >
> > Thank you again for your initial valuable feedback on our manuscript. While we understand that your time is very demanding, we are still waiting for your thoughts on our rebuttal materials (summarized in the "Author Rebuttal by Authors" section).
> >
> > Regarding all your concerns including the training speed, more ablation studies, single-stage training, etc., we believe they are all clearly addressed above.
> >
> > We would greatly appreciate any additional comments you could provide. Your time and consideration are highly valued.
> >
> > Regards,
> > Authors

---

### Author Rebuttal · Authors · 2023-08-10

We appreciate all insightful comments. After carefully improving the quality of our work, we present a document containing additional experimental results. And the reponses to the comments include:

- Clarification of our dual-ray visibility classifier.
- Evaluations of our method using sparse depth supervision.
- Comparisons of our method using different ray parameterizations.
- Comparisons of adopting our dual-ray visibility classifier to other baselines.
- Additional ablations on our dual-ray visibility classifier, post-processing, and the training strategy.
- Additional ablations on our ray-surface distance netowrk, including network architecture and thin structure reconstruction.
- Additional baseline comparisons.

---

### Decision · Program_Chairs · 2023-09-21

**Decision:**

Accept (poster)

**Comment:**

This paper received generally positive reviews. While there are clear limitations e.g. requirement of depth and long training time, the presented approach does allow accurate and efficient inference. Moreover, the technical contributions in the two-stage training of the ray-distance network are also interesting. On the balance, the AC would side with the reviewers recommending acceptance, but would encourage the authors to incorporate the reviewer suggestions to improve the presentation in the final version.